# A comprehensive model for the kyr and Myr time scales of Earth's axial magnetic dipole field

Matthias Morzfeld[1] and Bruce A. Buffett[2]

[1]Department of Mathematics, University of Arizona, 617 N. Santa Rita Ave., P.O. Box 210089, Tucson, Arizona 85721, USA
[2]University of California, Department of Earth & Planetary Science, 307 McCone Hall, Berkeley, CA 94720, USA

**Correspondence:** Matthias Morzfeld, mmo@math.arizona.edu

**Abstract.** We consider a stochastic differential equation model for Earth's axial magnetic dipole field. Our goal is to estimate the model's parameters using diverse and independent data sources that had previously been treated separately, so that the model is a valid representation of an expanded paleomagnetic record on kyr to Myr time scales. We formulate the estimation problem within the Bayesian framework and define a feature-based posterior distribution that describes probabilities of model parameters given a set of features, derived from the data. Numerically, we use Markov chain Monte Carlo (MCMC) to obtain a sample-based representation of the posterior distribution. The Bayesian problem formulation and its MCMC solution allow us to study the model's limitations and remaining posterior uncertainties. Our approach thus results in a reliable stochastic model for selected aspects of the long term behavior of the geomagnetic dipole field whose limitations and errors are well understood. We believe that such a model is useful for hypothesis testing and give a few examples of how the model can be used in this context. Another important aspect of our overall approach is that it can reveal inconsistencies between model and data, or within the various data sets. Identifying these shortcomings is a first and necessary step towards building more sophisticated models or towards resolving inconsistencies within the data.

## 1 Introduction

Earth possesses a time-varying magnetic field which is generated by the turbulent flow of liquid metal alloy in the core. The field can be approximated as a dipole with north and south magnetic poles slightly misaligned with the geographic poles. The dipole field changes over a wide range of timescales, from years to millions of years and these changes are documented by several different sources of data, see, e.g., Hulot et al. (2010). Satellite observations reveal changes of the dipole field over years to decades (Finlay et al., 2016), while changes on time scales of thousands of years are described by paleomagnetic data, including observations of the dipole field derived from archeological artifacts, young volcanics, and lacustrine sediments (Constable et al., 2016). Variations on even longer time scales of millions of years are recorded by marine sediments (Valet et al., 2005; Ziegler et al., 2011) and by magnetic anomalies in the oceanic crust (Ogg, 2012; Cande and Kent, 1995; Lowrie and Kent, 2004). On such long time scales, we can observe the intriguing feature of Earth's axial magnetic dipole field to reverse its polarity (magnetic north pole becomes the magnetic south pole and vice versa).

Understanding Earth's dipole field, at any time scale, is difficult because the underlying magnetohydrodynamic problem is highly nonlinear. For example, many numerical simulations are far from Earth-like due to severe computational constraints and more tractable mean-field models require questionable parameterizations. An alternative approach is to use "low-dimensional models" which aim at providing a simplified but meaningful representation of some aspects of Earth's geo-dynamo. Several such models have been proposed over the past years. The model of Gissinger (2012), for example, describes the Earth's dipole over millions of years by a set of three ordinary differential equations, one for the dipole, one for the non-dipole field and one for velocity variations at the core. A stochastic model for Earth's dipole over millions of years was proposed by Pétrélis et al. (2009). Other models have been derived by Rikitake (1958); Pétrélis and Fauve (2008).

Following Schmitt et al. (2001); Hoyng et al. (2001, 2002), we consider a stochastic differential equation (SDE) model for Earth's axial dipole. The basic idea is to model Earth's dipole field analogous to the motion of a particle in a double well potential. Time variations of the dipole field and dipole reversals then occur as follows. The state of the SDE is within one of the two wells of the double well potential and is pushed round by noise. The pushes and pulls by the noise process lead to variations of the dipole field around a typical value. Occasionally, however, the noise builds up to push the state over the potential well, which causes a change of its sign. A transition from one well to the other represents a reversal of Earth's dipole. The state of the SDE then remains, for a while, within the opposite well and the noise leads to time variations of the dipole field around the negative of the typical value. Then, the reverse of this process may occur.

A basic version of this model, which we call the "B13 model" for short, was discussed by Buffett et al. (2013). The drift and diffusion coefficients that define the B13 model are derived from the PADM2M data (Ziegler et al., 2011) which describe variations in the strengths of Earth's axial magnetic dipole field over the past 2 Myr. The PADM2M data are derived from marine sediments which means that the data are smoothed by sedimentation processes, see, e.g., Roberts and Winkhofer (2004). The B13 model, however, does not account directly for the effects of sedimentation. Buffett and Puranam (2017) try to mimic the effects of sedimentation by sending the solution of the SDE through a low-pass filter. With this extension, the B13 model is more suitable to be compared to the data record of Earth's dipole field on a Myr time-scale.

A basic assumption of an SDE model is that the noise process within the SDE is uncorrelated in time. This assumption is reasonable when describing the dipole field on the Myr time scale but is not valid on a shorter time scale of thousands of years. Buffett and Matsui (2015) derived an extension of the B13 model to extend it to time scales of thousands of years, by adding a time-correlated noise process. An extension of B13 to represent changes in reversal *rates* over the past 150 Myrs is considered by Morzfeld et al. (2018). Its use for predicting the probability of an imminent reversal of Earth's dipole is described by Morzfeld et al. (2017) and by Buffett and Davis (2018). The B13 model is also discussed by Meduri and Wicht (2016); Buffett et al. (2014); Buffett (2015).

The B13 model and its extensions are constructed with several data sets in mind that document Earth's axial dipole field over the kyr and Myr time scales. The data, however, are not considered simultaneously: the B13 model is based on one data source (paleomagnetic data on the Myr time scale) and some of its modifications are based on other data sources (the shorter record over the past 10 kyrs). Our goal is to construct a comprehensive model for Earth's axial dipole field by calibrating the B13 model to several independent data sources *simultaneously*, including

(i) observations of the strength of the dipole over the past 2 Myrs as documented by the PADM2M and Sint-2000 data sets (Ziegler et al., 2011; Valet et al., 2005);

(ii) observations of the dipole over the past 10 kyr as documented by CALS10k.2 (Constable et al., 2016);

(iii) reversals and reversal rates derived from magnetic anomalies in the oceanic crust (Ogg, 2012).

The approach ultimately leads to a family of SDE models, valid over Myr and kyr time scales, whose parameters are informed by a comprehensive paleomagnetic record composed of the above three sources of data. The results we obtain here are thus markedly different from previous work were data at different time scales are considered separately. We also use our framework to assess the effects of the various data sources on parameter estimates and to discover inconsistencies between model and data.

At the core of our model calibration is the Bayesian paradigm in which uncertainties in data are converted into uncertainties in model parameters. The basic idea is to merge prior information about the model and its parameters, represented by a prior distribution, with new information from data, represented by a likelihood, see, e.g., Reich and Cotter (2015); Asch et al. (2017). Priors are often assumed to be "uninformative", i.e., only conservative bounds for all parameters are known, and likelihoods describe point-wise model-data mismatch. Such likelihoods, however, are difficult to use when a variety of diverse data sources

are to be combined. For example, a point-wise mismatch of model and data is difficult to enforce when two different data sets report two different values for the same quantity (see, e.g., Figure 1). Assumed error models in the data can control the effects each data set has on the parameter estimates. Since error models describe what "*we do not know*", good error models are notoriously difficult to come by. In this context, we discover that the "shortness of the paleomagnetic record," i.e., the limited amount of data available, is the main source of uncertainty. For example, PADM2M or Sint-2000 provide a time series of 2,000

consecutive "data points" (2 Myr sampled once per kyr). Errors in power spectral densities, computed from such a short time series, dominate the expected errors in these data. Similarly, errors in the reversal rate statistics are likely dominated by the fact that only a small number of reversals, e.g., those that occurred over the past 30 Myr, are useful for computing reversal rates. Reliable error models should thus reflect errors due to the shortness of the paleomagnetic record, rather than building error models on assumed errors in the data.

To address these issues we substitute likelihoods based on point-wise mismatch of model and data by a "feature-based" likelihood, as discussed by Maclean et al. (2017); Morzfeld et al. (2018). A feature-based likelihood is based on error in "features" extracted from model outputs and data rather than the usual point-wise error. The feature-based approach enables unifying contributions from several independent data sources in a well-defined sense even if the various data may not be entirely self-consistent and further allows us to construct error models that reflect uncertainties induced by the shortness of the

paleomagnetic record. In addition, we perform a suite of numerical experiments to check, in hind-sight, our a priori assumptions about the error models.

## 2  Description of the data

Variations in the virtual axial dipole moment (VADM) over the past 2 Myr can be derived from stacks of marine sediment.Two different compilations are considered in this study: Sint-2000, (Valet et al., 2005) and PAMD2M (Ziegler et al., 2011). Both

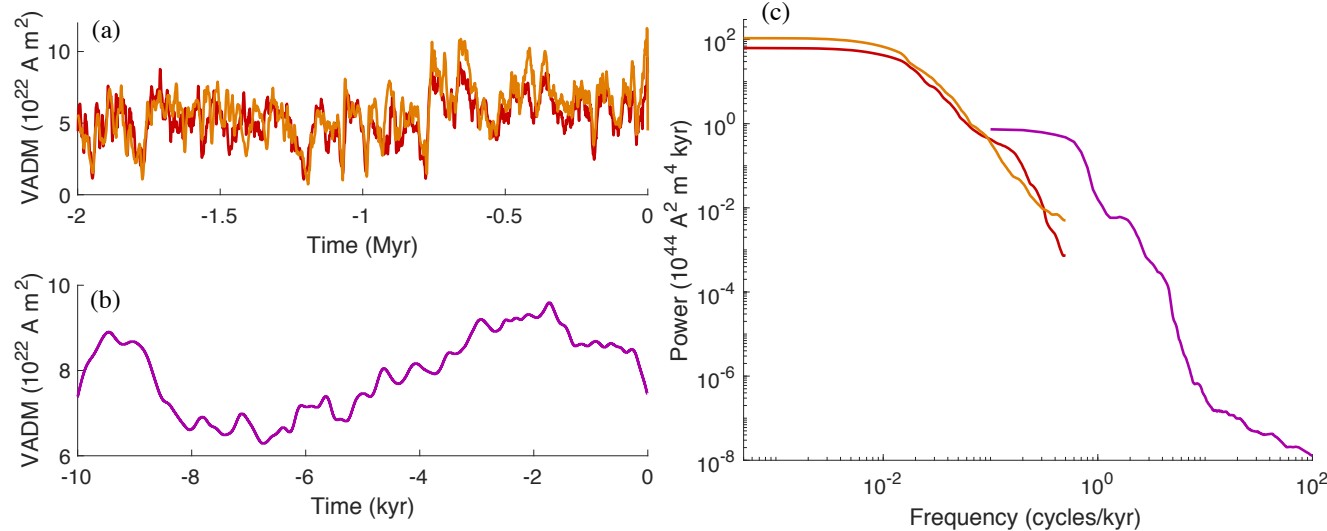

**Figure 1.** Data used in this paper. (a) Sint-2000 (orange) and PADM2M (red): VADM as a function of time over the past 2 Myr. (b) CALS10k.2: VADM as a function of time over the past 10 kyr. (c) Power spectral densities of the data in (a) and (b), computed by the multi-taper spectral estimation technique of Constable and Johnson (2005). Orange: Sint-2000. Red: PADM2M. Purple: CALS10k.2

of these data sets are sampled every 1 kyr and, thus, provide a time series of 2,000 consecutive VADM values. The PADM2M and Sint-2000 data sets are shown in the upper left panel of Figure 1. The CALS10k.2 data set, plotted in the lower left panel of Figure 1, describes variations of VADM over the past 10 kyr (Constable et al., 2016). The time dependence of CALS10k2 is represented using B-splines, so that the model can be sampled at arbitrary time intervals. We sample CALS10k.2 at an interval

5 of one year, although the resolution of CALS10k.2 is nominally 100 years (Constable et al., 2016).

Below we use features derived from power spectral densities (PSD) of the Sint-2000, PADM2M and CALS10k.2 data. The PSDs are computed by the multi-taper spectral estimation technique of Constable and Johnson (2005). A restricted range of frequencies are retained in the estimation to account for data resolution and other complications (see below). We show the resulting PSDs of the three data sets in the right panel of Figure 1. During parameter estimation we further make use of the

10 time average VADM and the standard deviation of VADM over time of the Sint-2000 and PADM2M data sets, listed in Table 1.

| Data source | Time avg. VADM ($10^{22}$ Am$^2$) | Std. dev. of VADM ($10^{22}$ Am$^2$) |
|---|---|---|
| PADM2M | 5.23 | 1.48 |
| Sint-2000 | 5.81 | 1.84 |

**Table 1.** Time average VADM and VADM standard deviation of PADM2M and Sint-2000.

Lastly, we make use of reversal rates of the Earth's dipole computed from the geomagnetic polarity time scale (Cande and Kent, 1995; Lowrie and Kent, 2004; Ogg, 2012). Using the chronology of Ogg (2012), we compute reversal rates for 5 Myr intervals from today up to 30 Myr ago. That is, we compute the reversal rates for the intervals $0\,\mathrm{Myr} - 5\,\mathrm{Myr}, 5\,\mathrm{Myr} - 10\,\mathrm{Myr}, \ldots, 25\,\mathrm{Myr} - 30\,\mathrm{Myr}$. This leads to the average reversal rate and standard deviation listed in Table 2. Increasing the

| Interval length | Average reversal rate (reversals/Myr) | Std. dev. (reversals/Myr) |
|---|---|---|
| 5 Myr | 4.23 | 1.01 |
| 10 Myr | 4.23 | 0.49 |

**Table 2.** Average reversal rate and standard deviation computed over the past 30 Myr using the chronology of Ogg (2012).

interval to 10 Myr leads to the same mean but decreases the standard deviation (see Table 2).

Note that the various data are not all consistent. For example, visual inspection of VADM (Figure 1), as well as comparison of the time average and standard deviation (Table 1) indicate that the PADM2M and Sint-2000 data sets report different VADM. These differences can be attributed, at least in part, to differences in the calibration of the marine sediment measurements and to differences in the way the measurements are stacked to recover the dipole component of the field. There are also notable

differences between the PSDs from CALS10k.2 and those from the lower resolution data sets (SINT-2000 and PADM2M) at the overlapping frequencies. Dating uncertainties, smoothing due to sedimentary processes and the finite duration of the records all contribute to these discrepancies. We do not attempt to identify the source of these discrepancies. Instead, we seek to recover parameter values for a stochastic model by combining a feature-based approach with realistic estimates of the data uncertainty (see Section 4).

We further note that the amount of data is rather limited: we have 2 Myrs of VADM sampled at 1/ kyr, 10 kyr of high frequency VADM and use a 30 Myr record to compute reversal rates. The limited amount of data directly affects how the accuracy of the data should be interpreted. As an example, the mean and standard deviation of the reversal rate, based on a 30 Myr record may not be accurate; errors in the PSDs of PADM2M, Sint-2000 or CALS10k.2 are dominated by the fact that these are computed from relatively short time series. We address these issues by using the feature-based approach that allows us

to build error models that reflect uncertainties due to the shortness of the paleomagnetic record. We further perform extensive numerical tests that allow us to check, in hind-sight, the validity our assumptions about errors (see Section 6).

## 3   Description of the model

Our models for variations in the dipole moment on Myr and kyr time scales are based on a scalar stochastic differential equation (SDE)

$$\mathrm{d}x = v(x)\mathrm{d}t + \sqrt{2D(x)}\mathrm{d}W, \tag{1}$$

where $t$ is time and where $x$ represents the VADM and polarity of the dipole, see, e.g., Schmitt et al. (2001); Hoyng et al. (2001, 2002); Buffett et al. (2013). A negative sign of $x(t)$ corresponds to the current polarity, a positive sign means reversed polarity. $W$ is Brownian motion, a stochastic process with the following properties: $W(0) = 0$, $W(t) - W(t + \Delta T) \sim \mathcal{N}(0, \Delta t)$, $W(t)$ is almost surely continuous for all $t \geq 0$, see, e.g., Chorin and Hald (2013). Here and below, $\mathcal{N}(\mu, \sigma^2)$ denotes a Gaussian random variable with mean $\mu$, standard deviation $\sigma$ and variance $\sigma^2$. Throughout this paper, we assume that the diffusion, $D(x)$, is constant, i.e., $D(x) = D$. Modest variations in $D$ have been reported on the basis of geodynamo simulations (Buffett and Matsui, 2015; Meduri and Wicht, 2016). Representative variations in $D$, however, have a small influence on the statistical properties of solutions of the SDE (1), see Buffett and Matsui (2015).

The function $v$ is called the "drift" and is derived from a double-well potential, $U'(x) = -v(x)$. Here, we consider drift coefficients of the form

$$v(x) = \gamma \frac{x}{\bar{x}} \cdot \begin{cases} (\bar{x} - x), & \text{if } x \geq 0 \\ (x + \bar{x}), & \text{if } x < 0 \end{cases}, \tag{2}$$

where $\bar{x}$ and $\gamma$ are parameters. The parameter $\bar{x}$ defines where the drift coefficient vanishes and also corresponds to the time average of the associated linear model

$$\mathrm{d}x^l = -\gamma(x^l - \bar{x})\mathrm{d}t + \sqrt{D}\mathrm{d}W, \tag{3}$$

which is obtained by Taylor expanding $v(x)$ at $\bar{x}$. It is now clear that the parameter $\gamma$ defines a relaxation time.

Nominal values of the parameters $\bar{x}$, $\gamma$ and $D$ are listed in Table 3. With the nominal values, the model exhibits "dipole

| | $\bar{x}$ | $D$ | $\gamma$ | $T_s$ | $a$ |
| | ($10^{22}$ Am$^2$) | ($10^{44}$ A$^2$m$^4$ kyr$^{-1}$) | (kyr$^{-1}$) | (kyr) | (kyr$^{-1}$) |
|---|---|---|---|---|---|
| Nominal value: | 5.23 | 0.3403 | 0.075 | 2.5 | 5 |
| Lower bound: | 0 | 0.0615 | 0.0205 | 1 | 5 |
| Upper bound: | 10 | 2.1 | 0.7 | 5 | 40 |

**Table 3.** Nominal parameter values and parameter bounds

reversals", which are represented by a change of the sign of $x$. This is the "basic" B13 model.

For computations, we discretize the SDE using a 4th order Runge-Kutta (RK4) method for the drift and an Euler-Maruyama method for the diffusion. This results in the discrete time B13 model

$$x_k = f(x_{k-1}, \Delta t) + \sqrt{2D\Delta t}\, w_k, \quad w_k \sim \mathcal{N}(0, 1), \text{ iid} \tag{4}$$

where $\Delta t$ is the time step, $\sqrt{\Delta t}\, w_k$ is the discretization of Brownian motion $W$ in (1) and where $f(x_{k-1}, \Delta t)$ is the RK4 step. Here, iid stands for "independent and identically distributed", i.e., each random variable $w_k$, for $k > 0$, has the same Gaussian probability distribution, $\mathcal{N}(0, 1)$, and $w_i$ and $w_j$ are independent for all $i \neq j$. We distinguish between variations in the Earth's dipole over kyr to Myr time scales and, for that reason, present modifications of the basic B13 model (4).

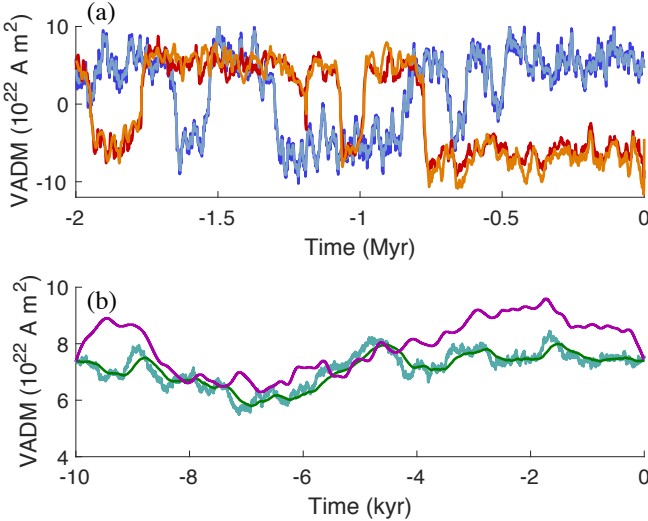

**Figure 2.** Simulations with nominal parameter values and data on the Myr and kyr scales. (a) VADM as a function of time on the Myr time scale. The output of the Myr model, $x_j^{\text{Myr}}$, is shown in dark blue (often hidden). The smoothed output, $x_j^{\text{Myr,s}}$, is shown in light blue. The signed Sint-2000 and PADM2M are shown in orange and red with signs (reversal timings) taken from Cande and Kent (1995). (b) VADM as a function of time on the kyr time scale. The output of the kyr-model with uncorrelated noise is shown in turquoise. The output of the kyr-model with correlated noise is shown in green. VADM of CALS10k.2 is shown in purple.

### 3.1 Models for the Myr amd kyr time scale

For simulations over Myr time scales we chose a time-step $\Delta t = 1$ kyr, corresponding to the sampling time of the Sint-2000 and PADM2M data. On a Myr time scale, the primary source of paleomagnetic data in Sint-2000 and PADM2M are affected by gradual acquisition of magnetization due to sedimentation processes, which amount to an averaging over a (short) time
interval, see, e.g., Roberts and Winkhofer (2004). We follow Buffett and Puranam (2017) and include the smoothing effects of sedimentation in the model by convolving the solution of (4) by a Gaussian filter

$$g(t) = \sqrt{\frac{6}{\pi T_s^2}} \cdot \exp\left(-\frac{6t^2}{T_s^2}\right), \tag{5}$$

where $T_s$ defines the duration of smoothing, i.e., the width of a time window over which we average. The nominal value for $T_s$ is given in Table 3. The result is a smoothed Myr model whose state is denoted by $x^{\text{Myr,s}}$. Simulations with the "Myr model"
and the nominal parameters of Table 3 are shown in Figure 2(a) where we plot the model output $x_j^{\text{Myr}}$ in dark blue and the smoothed model output, $x_j^{\text{Myr,s}}$, in a lighter blue over a period of 2 Myr.

On a Myr scale, the assumption that the noise is uncorrelated in time is reasonable because one focuses on low frequencies and large sample intervals of the dipole, as in Sint-2000 and PADM2M, whose sampling interval is 1/kyr. On a shorter time scale, as in CALS10k.2, this assumption is not valid and a correlated noise is more appropriate (Buffett and Matsui, 2015).
Computationally, this means that we swap the uncorrelated, iid, noise in (4) for a noise that has a short but finite correlation

time. This can be done by "filtering" Brownian motion. The resulting discrete time model for the kyr time scale is

$$y_k = (1 - a\Delta t)y_{k-1} + \sqrt{2a\Delta t}\, w_k, \quad w_k \sim \mathcal{N}(0,1), \text{ iid} \tag{6}$$

$$x_k = f(x_{k-1}, \Delta t) + \sqrt{Da}\,\Delta t\, y_k, \tag{7}$$

where $a$ is the model parameter that defines the correlation time $T_c = 1/a$ of the noise and $\Delta t = 1$ yr. A 10 kyr simulation of
the kyr models with uncorrelated and correlated noise using the nominal parameters of Table 3, are shown in Figure 2(b) along
with the CALS10k.2 data.

## 3.2 Approximate power spectral densities

Accurate computation of the power spectral density (PSD) from the time-domain solution of the B13 model requires extremely
long simulations. For example, the PSDs of two (independent) 1 billion year simulations with the Myr model are still surpris-
ingly different. In fact, errors that arise due to "short" simulations substantially outweigh errors due to linearization. Recall that
the PSD of the linear model (3) is easily calculated to be

$$\hat{x}^l(f) = \frac{2D}{\gamma^2 + 4\pi^2 f^2}, \tag{8}$$

where $f$ is the frequency (in $1/\text{kyr}$). Since the Fourier transform of the Gaussian filter is known analytically, the PSD of the
smoothed linear model is also easy to calculate:

$$\hat{x}^{l,s}(f) = \frac{2D}{\gamma^2 + 4\pi^2 f^2} \cdot \exp\left(-\frac{4\pi^2 f^2 T_s^2}{12}\right). \tag{9}$$

Similarly, an analytic expression for the PSD of the kyr-model with correlated noise in equations (6)-(7) can be obtained by
taking the limit of continuous time ($\Delta t \to 0$):

$$\hat{x}^{l,kyr}(f) = \frac{2D}{\gamma^2 + 4\pi^2 f^2} \cdot \frac{a^2}{a^2 + 4\pi^2 f^2}. \tag{10}$$

Here, the first term is as in (8) and the second term appears because of the correlated noise.

Figure 3 illustrates a comparison of the PSDs obtained from simulations of the nonlinear models and their linear approxima-
tions. Specifically, the PSDs of the (smoothed) Myr scale nonlinear model, computed from a 50 Myr simulation, are shown in
comparison to the approximate PSDs in equations (8)-(9). Note that the PSD of the smoothed model output, $x^{\text{Myr,s}}$, taking into
account sedimentation processes, rolls-off quicker than the PSD of $x_j^{\text{Myr}}$. For that reason, the PSD of the smoothed model seems
to fit the PSDs of the Sint-2000 and PADM2M data "better", i.e., we observe a similarly quick roll-off at high frequencies in
model and data; see also Buffett and Puranam (2017). The PSD of the kyr model with correlated noise, computed from a 10
kyr simulation, is also shown in Figure 3 in comparison with the linear PSD in equation (10). The good agreement between the
theoretical spectra of the linear models and the spectra of the nonlinear models justifies the use of the linear approximation.
We have further noted in numerical experiments that the agreement between the nonlinear and linear spectra increases with
increasing simulation time, however a "perfect" match requires extremely long simulations of the nonlinear model (hundreds

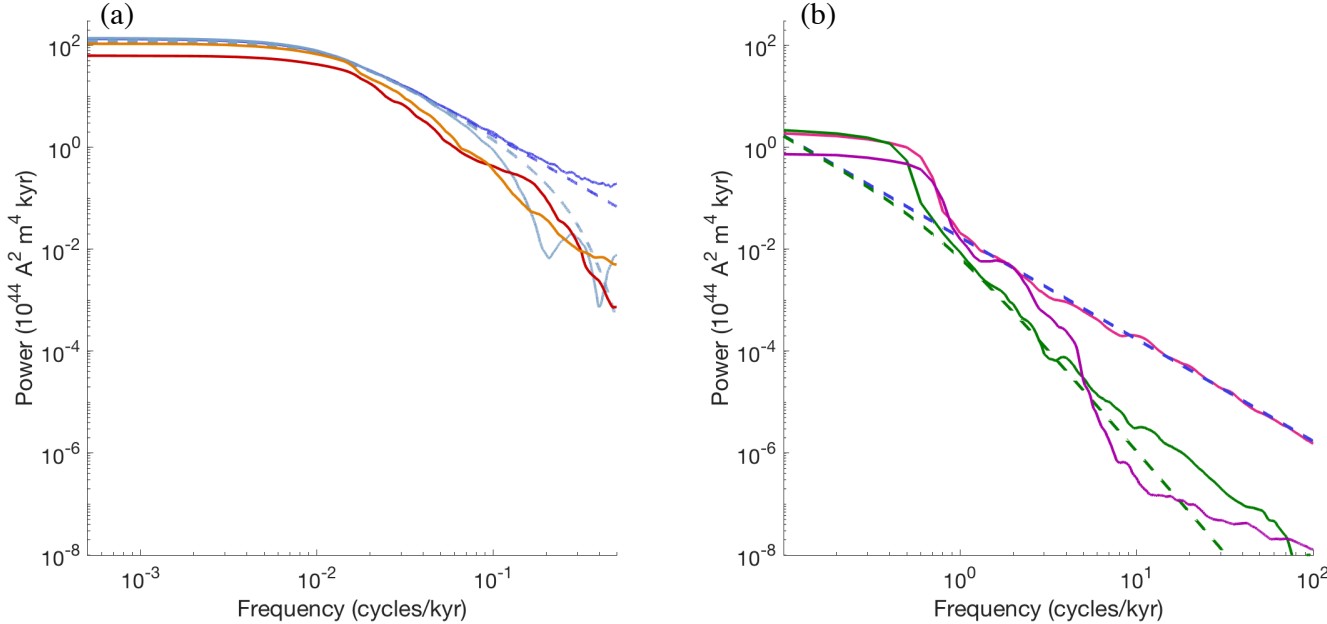

**Figure 3.** Power spectral densities of the model with nominal parameter values. (a) *Myr-model*: A PSD of a 50 Myr simulation with the Myr model is shown as a solid dark blue line. The corresponding theoretical PSD of the linear model is shown as a dashed dark blue line. A PSD of a 50 Myr simulation with the Myr model and high-frequency roll-off is shown as a solid light blue line. The corresponding theoretical PSD of the linear model is shown as a dashed light blue line. The PDSs of Sint-2000 and PADM2M are shown in orange and red. (b) *kyr model*: A PSD of a 10 kyr simulation of the kyr model with uncorrelated noise is shown as a solid pink line. The corresponding theoretical spectrum of the linear model is shown as a dashed blue line. A PSD of a 10 kyr simulation of the kyr model with correlated noise is shown as a solid green line. The corresponding theoretical spectrum is shown as a dashed green line. The PDSs of CALS10k.2 is shown in purple. All PSDs are computed by the multi-taper spectral estimation technique of Constable and Johnson (2005).

of billions of years). The approximate PSDs, based on the linear models, will prove useful in the construction of likelihoods in Section 4.2.

In addition to a good match of the PSDs of the nonlinear and linear models, we note that the PSDs of the model match, at least to some extent, the PSDs of the data (Sint-2000, PADM2M and CALS10k.2). This means that our choice for the nominal values is "reasonable" because this choice leads to a reasonable fit to the data. The goal of using a Bayesian approach to parameter estimation, described in Section 4, is to improve this fit and to equip the (nominal) parameter values with an error estimate, i.e., to define and compute a distribution over the model parameters. This will lead to an improved fit, along with an improved understanding of model uncertainties.

### 3.3 Approximate reversal rate, VADM time average and VADM standard deviation

The nonlinear SDE model (1) and its discretization (4) exhibit reversals, i.e., a change in the sign of $x$. Moreover, the overall "power", i.e., the area under the PSD curve, is given by the standard deviation of the absolute value of $x(t)$ over time. Another important quantity of interest is the time averaged value of the absolute value of $x(t)$, which describes the average strength of the dipole field. In principle, these quantities (reversal rate, time average and standard deviation) can be computed from simulations of the Myr and kyr model in the time domain. Similarly to what we found in the context of PSD computations and approximations, we find that estimates of the reversal rate, time average and standard deviation are subject to large errors unless the simulation time is very long (hundreds of millions of years). Using the linear model and Kramers formula, however, one can approximate the time average, reversal rate and standard deviation without simulating the nonlinear model (see below). Computing the approximate values is instantaneous (evaluation of simple formulas) and the approximations are comparable to what we obtain from very long simulations with the nonlinear model. As is the case with the PSD approximations based on linear models, the below approximations of the reversal rate, time average and standard deviation based will prove useful for formulating likelihoods in Section 4.2.

Specifically, the parameter $\bar{x}$ defines the time average of the linear model (3) and it also defines where the drift term (2) vanishes. These values coincide quite closely with the time average of the nonlinear model which suggests the approximation

$$E(x) \approx \bar{x} \cdot 10^{22} \text{ Am}^2. \tag{11}$$

The reversal rate can approximated by Kramers formula (Buffett and Puranam, 2017; Risken, 1996)

$$r \approx \frac{\gamma}{2\pi} \cdot \exp\left(-\frac{\gamma \bar{x}^2}{6D}\right) \cdot 10^3 \text{ Myr}^{-1}. \tag{12}$$

The standard deviation is the square root of the area under the PSD. Using the linear model that incorporates the effects of smoothing (due to sedimentation), one can approximate the standard deviation by computing the integral of the PSD in equation (9):

$$\sigma \approx \left(\frac{D}{\gamma} \exp\left(\frac{(\gamma T_s)^2}{12}\right) \text{erfc}\left(\frac{\gamma T_s}{2\sqrt{3}}\right)\right)^{1/2} \cdot 10^{22} \text{ Am}^2, \tag{13}$$

where erfc$(\cdot)$ is the (Gauss) error function. Without incorporating the smoothing, the standard deviation based on the linear model would be the integral of the PDS in (8), which is $\sigma \approx \sqrt{D/\gamma}$. The exponential and error function terms in (13) can thus be interpreted as a correction factor that accounts for the effects of sedimentation. It is easy to check that this correction factor is always smaller than one, i.e., the (approximate) standard deviation accounting for sedimentation effects is smaller than the (approximate) standard deviation that does not account for these effects.

For the nominal parameter values in Table 3 we calculate a time average of $\bar{x} = 5.23 \cdot 10^{22} \text{ Am}^2$, a standard deviation of $\sigma \approx 2.07 \cdot 10^{22} \text{ Am}^2$ and a reversal rate $r \approx 4.37 \text{ Myr}^{-1}$. These should be compared to the corresponding values of PADM2M and Sint-2000 in Table 1 and to the reversal rate from the geomagnetic polarity time scale in Table 2. Similar to what we observed of the model-data fit in terms of (approximate) PSDs, we find that the nominal parameter values lead to a "reasonable"

fit of the model's reversal rate, time average and standard deviation. The Bayesian parameter estimation in Section 4 will improve this fit and lead to a better understanding of model uncertainties.

## 3.4 Parameter bounds

The Bayesian parameter estimation, described in Section 4, makes use of "prior" information about the model parameters. We formulate prior information in terms of parameter bounds and construct uniform prior distributions with these bounds. The parameter bounds we use are quite wide, i.e., the upper bounds are probably too large and the lower bound are probably too small, but this is not critical for our purposes as we explain in more detail in Section 4.

The parameter $\gamma$ is defined by the inverse of the dipole decay time (Buffett et al., 2013). An upper bound on the dipole decay time $\tau_{\mathrm{dec}}$ is given by the slowest decay mode $\tau_{\mathrm{dec}} \leq R^2/(\pi^2 \eta)$, where $R$ is the radius of the Earth and $\eta = 0.8\,\mathrm{m}^2/\mathrm{s}$ is the magnetic diffusivity. Thus, $\tau_{\mathrm{dec}} \leq 48.6$ kyr, which means that $\gamma \geq 0.0205\ \mathrm{kyr}^{-1}$. This is a fairly strict lower bound because the dipole may relax on timescales shorter than the slowest decay mode and a recent theoretical calculation (Pourovskii et al., 2017) suggests that the magnetic diffusivity may be slightly larger than $0.8\mathrm{m}^2/\mathrm{s}$. Both of these changes would cause the lower bound for $\gamma$ to increase. To obtain an upper bound for $\gamma$, we note that if $\gamma$ is large, the magnetic decay is short, which means that it becomes increasingly difficult for convection in the core to maintain the magnetic field. The ratio of dipole decay time $\tau_{\mathrm{dec}}$ to advection time $\tau_{\mathrm{adv}} = L/V$, where $L = 2259$ km is the width of the fluid shell and $V = 0.5$ mm/s, needs to be 10:1 or (much) larger. This leads to the upper bound $\gamma \leq 0.7\ \mathrm{kyr}^{-1}$.

Bounds for the parameter $D$ can be found by considering the linear Myr time scale model in equation (3), which suggests that the variance of the dipole moment is $\mathrm{var}(x) = D/\gamma$, see also Buffett et al. (2013). Thus, we may require that $D \sim \mathrm{var}(x)\gamma$. The average of the variance of Sint-2000 ($\mathrm{var}(x) = 3.37 \cdot 10^{44}\ \mathrm{A}^2\mathrm{m}^4$) and PADM2M ($\mathrm{var}(x) = 2.19 \cdot 10^{44}\ \mathrm{A}^2\mathrm{m}^4$) is $\mathrm{var}(x) \approx 2.78 \cdot 10^{44}\ \mathrm{A}^2\mathrm{m}^4$. We use the rounded up value $\mathrm{var}(x) \approx 3 \cdot 10^{44}\ \mathrm{A}^2\mathrm{m}^4$ and, together with the lower and upper bounds on $\gamma$, this leads to the lower and upper bounds $0.062 \cdot 10^{44}\mathrm{A}^2\mathrm{m}^4\mathrm{kyr}^{-1} \leq D \leq 2.1 \cdot 10^{44}\mathrm{A}^2\mathrm{m}^4\mathrm{kyr}^{-1}$.

The smoothing time, $T_s$, due to sedimentation and the correlation parameter for the noise, $a$, define the roll-off frequency of the power spectra for the Myr and kyr models, respectively. We assume that $T_s$ is within the interval $[1,5]$ kyr, and that the correlation time $a^{-1}$ is within $[0.025, 0.2]$ kyr (i.e. $a$ within [5, 40] $\mathrm{kyr}^{-1}$). These choices enforce that $T_s$ controls roll-off at lower frequencies (Myr model) and $a$ controls the roll-off at higher frequencies (kyr model). Bounds for the parameter $\bar{x}$ are not easy to come by and we assume wide bounds, $\bar{x} \in [0,10] \cdot 10^{22}\ \mathrm{Am}^2$. Here, $\bar{x} = 0$ is the lowest lower bound we can think of since the average value of the field is always normalized to be positive. The value of the upper bound of $\bar{x} \leq 10$ is chosen to be excessively large – the average field strength over the last 2 Myr is $\bar{x} \approx 5$. Lower and upper bounds for all five model parameters are summarized in Table 3.

## 4   Formulation of the Bayesian parameter estimation problem and numerical solution

The family of models, describing kyr and Myr time-scales and accounting for sedimentation processes and correlations in the noise process, has five unknown parameters, $\bar{x}, D, \gamma, T_s, a$. We summarize the unknown parameters in a "parameter vector"

$\theta = (\bar{x}, D, \gamma, T_s, a)^T$. Our goal is to estimate the parameter vector $\theta$ using a Bayesian approach, i.e., to sharpen prior knowledge about the parameters by using the data described in Section 2. This is done by expressing prior information about the parameters in a prior probability distribution $p_0(\theta)$, and by defining a likelihood $p_l(y|\theta)$, $y$ being shorthand notation for the data of Section 2. The prior distribution describes information we have about the parameters independently of the data. The likelihood describes the probability of the data given the parameters $\theta$ and, therefore, connects model output and data. The prior and likelihood define the posterior distribution

$$p(\theta|y) \propto p_0(\theta)p_l(y|\theta). \tag{14}$$

The posterior distribution combines the prior information with the information we extract from the data. In particular, we can estimate parameters based on the posterior distribution. For example, we can compute the posterior mean and posterior standard deviation for the various parameters and we can also compute correlations between the parameters. The posterior distribution contains all information we have about the model parameters, given prior knowledge and information extracted from the data. Thus, the SDE model with random parameters, whose distribution is the posterior distribution, represents a comprehensive model of the Earth's dipole in view of the data we use.

On the other hand, the posterior distribution depends on several assumptions: since we define the prior and likelihood, we also implicitly define the posterior distribution. In particular, formulations of the likelihood require that one be able to describe anticipated errors in the data as well as anticipated model error. Such error models are difficult to come by in general, but even more so when the amount of data is limited. We address this issue by first formulating "reasonable" error models, followed by a set of numerical tests that confirm (or disprove) our choices of error models (see Section 6). In our formulation of error models, we focus on errors that arise due to the shortness of the paleomagnetic record because these errors dominate.

We solve the Bayesian parameter estimation problem numerically by using a Markov chain Monte Carlo (MCMC) method. An MCMC method generates a (Markov) chain of parameter values whose stationary distribution is the posterior distribution. The chain is constructed by proposing a new parameter vector and then accepting or rejecting this proposal with a specified probability that takes the posterior probability of the proposed parameter vector into account. A numerical solution via MCMC thus requires that the likelihood be evaluated for every proposed parameter vector. Below we formulate a likelihood that involves computing the PSDs of the Myr and kyr model, as well as reversal rates, time averages and standard deviations. As explained in Section 3, obtaining these quantities from simulations with the nonlinear models requires extremely long simulations. Long simulations, however, require more substantial computations. This perhaps would not be an issue if we were to compute the PDSs, reversal rate and other quantities once, but the MCMC approach we take requires repeated computing. For example, we consider Markov chains of length $10^6$, which requires $10^6$ computations of PSDs, reversal rates etc. Moreover, we will repeat these computations in a variety of settings to assess the validity of our error models (see Section 6). To keep the computations feasible (fast), we thus decided to use the approximations of the PSDs, reversal rate, time average and standard deviation, based on the linear models (see Section 3), to define the likelihood. Evaluation of the likelihood is then instantaneous because simulations with the nonlinear models are replaced by formulas that are simple to evaluate. Using the approximation

is further justified by the fact that the approximate PDSs, reversal rates, time averages and standard are comparable to what we obtain from very long simulations with the nonlinear model.

## 4.1 Prior distribution

The prior distribution describes knowledge about the model parameters we have *before* we consider the data. In Section 3.4, we discussed lower and upper bounds for the model parameters and we use these bounds to construct the prior distribution. This can be achieved by assuming a uniform prior over a five-dimensional hyper-cube whose corners are defined by the parameter bounds in Table 1. Note that the bounds we derived in Section 3.4 are fairly wide. Wide bounds are preferable for our purposes, because wide bounds implement minimal prior knowledge about the parameters. With such "uninformative priors", the posterior distribution, which contains information from the data, reveals how well the parameter values are constrained by data. More specifically, if the uniform prior distribution is morphed into a posterior distribution that describes a well-defined "bump" of posterior probability mass in parameter space, then the model parameters are constrained by the data (to be within the bump of posterior probability "mass"). If the posterior distribution is nearly equal to the prior distribution, then the data have nearly no effect on the parameter estimates and, therefore, the data do not constrain the parameters.

## 4.2 Feature-based likelihoods

In "data assimilation" or, more generally, in Bayesian estimation, information from a mathematical or numerical model is merged with information from data (sometimes called "observations") by combining a prior distribution with a likelihood to define a posterior distribution. The likelihood describes the probability of the data given the numerical model and its parameters and, therefore, connects the model and the data. Likelihoods are often based on a point-wise mismatch of the model outputs and the data.

We wish to use to use a collection of paleomagnetic observations to calibrate and constrain all five model parameters. For this purpose, we use the data sets Sint-2000, PADM2M and CALS10k.2, as well as information about the reversal rate based on the geomagnetic polarity time scale (see Section 2). The various data sets are not consistent and, for example, Sint-2000 and PADM2M report different VADM values at the same time instant (see Figure 1). Likelihoods that are defined in terms of a point-wise mismatch of model and data balance the effects of each data set via (assumed) error covariances: the data set with smaller error covariances has a stronger effect on the parameter estimates. Accurate error models, however, are hard to come by. For this reason, we use an alternative approach called "feature-based data assimilation" (see Morzfeld et al. (2018); Maclean et al. (2017)). The idea is to extract "features" from the data and to subsequently define likelihoods that are based on the mismatch of the features derived from the data and the model. Below, we formulate features that account for discrepancies across the various data sources and derive error models for the features. The error models are built to reflect uncertainties that arise due to the shortness of the paleomagnetic record. The resulting feature-based posterior distribution describes the probability of model parameters in view of the features. Thus, model parameters with a large feature-based posterior probability lead to model features that are comparable to the features derived from the data, within the assumed uncertainties due to the shortness of the paleomagnetic record.

Specifically, we define likelihoods based on features derived from PSDs of the Sint-2000, PADM2M and CALS10k.2 data sets, as well as the reversal rate, time average VADM and VADM standard deviation. The overall likelihood consists of three factors:

(i) one factor corresponds to the contributions from the reversal rate, time average VADM and VADM standard deviation data, which we summarize as "time domain data" from now on for brevity;

(ii) one factor describes the contributions from data at low frequencies of $10^{-4} - 0.5$ cycles per kyr (PADM2M and Sint-2000);

(iii) one factor describes contribution of data at high frequencies of $0.9 - 9.9$ cycles/kyr (CALS10k.2)

In the Bayesian approach, and assuming that errors are independent, this means that the likelihood, $p_l(y|\theta)$ in equation (14) can be written as the product of three terms

$$p_l(y|\theta) \propto p_{l,\text{td}}(y|\theta)\, p_{l,\text{lf}}(y|\theta) p_{l,\text{hf}}(y|\theta), \tag{15}$$

where $p_{l,\text{td}}(y|\theta)$, $p_{l,\text{lf}}(y|\theta)$ and $p_{l,\text{hf}}(y|\theta)$ represent the contributions from the time domain data (reversal rate, time average VADM and VADM standard deviation), the low frequencies and the high frequencies; recall that $y$ is shorthand notation for all the data we use. We now describe how each component of the overall likelihood is constructed.

### 4.2.1 Reversal rates, time average VADM and VADM standard deviation

We define the likelihood component of the time domain data based on the equations

$$y_{\text{rr}} = h_{\text{rr}}(\theta) + \varepsilon_{\text{rr}}, \tag{16}$$

$$y_{\bar{x}} = h_{\bar{x}}(\theta) + \varepsilon_{\bar{x}}, \tag{17}$$

$$y_\sigma = h_\sigma(\theta) + \varepsilon_\sigma, \tag{18}$$

where $y_{\text{rr}}$, $y_{\bar{x}}$ $y_\sigma$ are features derived from the time domain data, $h_{\text{rr}}(\theta)$, $h_{\bar{x}}(\theta)$ and $h_\sigma(\theta)$ are functions that connect the model parameters to the features, based on the approximations described in Section 3.3, and where $\varepsilon_{\text{rr}}$, $\varepsilon_{\bar{x}}$ and $\varepsilon_\sigma$ are independent Gaussian error models with mean zero and variances $\sigma_{\text{rr}}^2$, $\sigma_{\bar{x}}^2$, and $\sigma_\sigma^2$. Taken all together, the likelihood term $p_{l,\text{td}}(y|\theta)$ in (15) is then given by the product of the three likelihoods defined by equations (16), (17) and (18):

$$p_{l,\text{td}}(y|\theta) \propto \exp\left(-\frac{1}{2}\left(\left(\frac{y_{\text{rr}} - h_{\text{rr}}(\theta)}{\sigma_{\text{rr}}}\right)^2 + \left(\frac{y_{\bar{x}} - h_{\bar{x}}(\theta)}{\sigma_{\bar{x}}}\right)^2 + \left(\frac{y_\sigma - h_\sigma(\theta)}{\sigma_\sigma}\right)^2\right)\right). \tag{19}$$

The reversal rate feature is simply the average reversal rate we computed from the chronology of Ogg (2012) (see Section 2), i.e., $y_{\text{rr}} = 4.23$ reversals/Myr. The function $h_{\text{rr}}(\theta)$ is based on the approximation using Kramers formula in equation (20):

$$h_{\text{rr}}(\theta) = \frac{\gamma}{2\pi} \exp\left(-\frac{\gamma \bar{x}^2}{6D}\right) \cdot 10^3 \text{ reversals/Myr}. \tag{20}$$

The time average feature is the mean of the time averages of PADM2M and Sint-2000: $y_{\bar{x}} = 5.56 \cdot 10^{22}$ Am$^2$. The function $h_{\bar{x}}(\theta)$ is based on the linear approximation discussed in Section 3.3, i.e., $h_{\bar{x}}(\theta) = \bar{x}$. The feature for the VADM standard

deviation is the average of the VADM standard deviations of PADM2M and Sint-2000: $y_\sigma = 1.66 \cdot 10^{22}$ Am$^2$. The function $h_\sigma(\theta)$ uses the linear approximation of the standard deviation (13):

$$h_\sigma(\theta) = \left( \frac{D}{\gamma} \exp\left( \frac{(\gamma T_s)^2}{12} \right) \operatorname{erfc}\left( \gamma T_s/2/\sqrt{(3)} \right) \right)^{1/2} \cdot 10^{22} \text{ Am}^2 \text{ kyr}^2 \tag{21}$$

Candidate values for these error variances are as follows. The error variance of the reversal rate, $\sigma_{\mathrm{rr}}^2$, can be based on the standard deviations we computed from the Ogg (2012) chronology in Table 2. Thus, we might use the standard deviation of the 10-Myr average and take $\sigma_{\mathrm{rr}} = 0.5$. One can also use the model with nominal parameter values (see Table 3) to compute candidate values of the standard deviation $\sigma_{\mathrm{rr}}$. We perform 1,000 independent 10 Myr simulations and, for each simulation, determine the reversal rate. The standard deviation of the reversal rate based on these simulations is $0.69$ reversal per Myr, which is comparable to the $0.5$ reversals per Myr we computed from the Ogg (2012) chronology, using an interval length of 10 Myr. Similarly, the standard deviation of the reversal rate of 1,000 independent 5 Myr simulations is $0.97$, which is also comparable to the standard deviation of $1.01$ reversals per Myr, suggested by the Ogg (2012) chronology, using an interval length of 5 Myr.

A candidate for the standard deviation of the time average VADM is the difference of the time averages of Sint-2000 and PADM2M, which gives $\sigma_{\bar{x}} = 0.48 \cdot 10^{22}$ Am$^2$. Similarly, one can define the standard deviation $\sigma_\sigma$ by the difference of VADM standard deviations (over time) derived from Sint-2000 and PADM2M. This gives $\sigma_\sigma = 0.36 \cdot 10^{22}$ Am$^2$. We can also derive error covariances using the model with nominal parameters and perform 1,000 independent 2 Myr simulations. For each simulation, we compute the time average and the VADM standard deviations, which then allows us to compute standard deviations of these quantities. Specifically, we found a value of $0.26 \cdot 10^{22}$ Am$^2$ for the standard deviation of the time average and $0.11 \cdot 10^{22}$ Am$^2$ for standard deviation of the VADM standard deviation. These values are comparable to what we obtained from the data, especially if we base the standard deviations on *half* of the difference of the PADM2M and Sint-2000 values, i.e., assuming that the data sets are within two standard deviations (rather than within one, which we assumed above).

A difficulty with these error covariances is that we have few time domain observations compared with the large number of spectral data in the power spectra (see below). This vast difference in the number of time domain and spectral data means that the spectral data can overwhelm the recovery of model parameters. To ensure that time domain observations also contribute to the parameter estimates we lower the error variances $\sigma_{\mathrm{rr}}$, $\sigma_{\bar{x}}$ and $\sigma_\sigma$, derived from the data by a factor of 100 and set

$$\sigma_{\mathrm{rr}} = 0.05 \text{ reversals/Myr}, \quad \sigma_{\bar{x}} = 0.048 \cdot 10^{22} \text{ Am}^2 \quad \sigma_\sigma = 0.036 \cdot 10^{22} \text{ Am}^2. \tag{22}$$

We discuss this choice and its consequences on parameter estimates and associated uncertainty in more detail in Section 6. An alternative approach would be to reduce the number of spectral data, e.g., by condensing the spectra into "a few" characteristics such as corner frequencies or slopes, see also (Bärenzung et al., 2018). The difficulty with such an approach, however, is that corner frequencies etc. are not easy to extract from the PDSs of the data. For these reasons, we pursue in this paper a "tuning" of the error variances to control the impact of the time domain data on the parameter estimates, but alternative strategies are straightforward to implement within our overall feature-based Bayesian estimation approach.

### 4.2.2 Low frequencies

The component $p_{l,\text{lf}}(y|\theta)$ of the feature-based likelihood (15) addresses the behavior of the dipole at low frequencies of $10^{-4} -$ 0.5 cycles per kyr and is based on the PSDs of the Sint-2000 and PADM2M data sets. We construct the likelihood using the equation

$$y_{\text{lf}} = h_{\text{lf}}(\theta) + \varepsilon_{\text{lf}}, \tag{23}$$

where $y_{\text{lf}}$ is a feature that represents the PSD of the Earth's dipole field at low frequencies, $h_{\text{lf}}(\theta)$ maps the model parameters to the data $y_{\text{lf}}$ and where $\varepsilon_{\text{lf}}$ represents the errors we expect.

We define $y_{\text{lf}}$ to be the mean of the PSDs of Sint-2000 and PADM2M. The function $h_{\text{lf}}(\theta)$ maps the model parameters to the feature $y_{\text{lf}}$ and is based on the PSD of the linear model (3). To account for the smoothing introduced by sedimentation

processes we define $h_{\text{lf}}(\theta)$ to be a function that computes the PSD of the Myr model by using the "un-smoothed" spectrum of equation (8) for frequencies less than 0.05 cycles/kyr, and uses the "smoothed" spectrum of equation (9) for frequencies between $0.05 - 0.5$ cycles/kyr:

$$h_{\text{lf}}(\theta) = \frac{2D}{\gamma^2 + 4\pi^2 f^2} \cdot \begin{cases} 1 & \text{if } f \leq 0.05 \\ \exp\left(-(4\pi^2 f^2 T_s^2)/12\right) & \text{if } 0.05 < f \leq 0.5 \end{cases} \tag{24}$$

Note that $h_{\text{lf}}(\theta)$ does not depend on $\bar{x}$ or $a$. This also means that the data regarding low frequencies are not useful for deter-

mining these two parameters (see Section 6).

The uncertainty introduced by sampling the VADM once per kyr for only 2 Myrs is the dominant source of error in the power spectral densities. For a Gaussian error model $\varepsilon_{\text{lf}}$ with zero mean, this means that the error covariance should describe uncertainties that are induced by the limited amount of data. We construct such a covariance as follows. We perform $10^4$ simulations, each of 2 Myr, with the nonlinear Myr model (4) and its nominal parameters (see Table 1). We compute the PSD

of each simulation and build the covariance matrix of the $10^4$ PSDs. In the left panel of Figure 4 we illustrate the error model by plotting the PSDs of PADM2M (red), Sint-2000 (orange), their mean, $y_{\text{lf}}$, (dark blue), and $5 \cdot 10^3$ samples of $\varepsilon_{\text{lf}}$ added to $y_{\text{lf}}$ (grey). Since the PSDs of Sint-2000 and PADM2M are well within the cloud of PSDs we generated with the error model, this choice for modeling the expected errors in low frequency PSDs seems reasonable to us.

For comparison, we also plot $10^3$ samples of an error model that only accounts for the reported errors in Sint-2000. This

is done by adding independent Gaussian noise, whose standard deviation is given by the Sint-2000 data set every kyr, to the VADM of Sint-2000 and PADM2M. This results in $10^3$ "perturbed" versions of Sint-2000 or PADM2M. For each one, we compute the PSD and plot the result in the right panel of Figure 4. The resulting errors are smaller than the errors induced by the shortness of the record. In fact, the reported error does not account for the difference in the Sint-2000 and PADM2M data sets. This suggests that the reported error is too small.

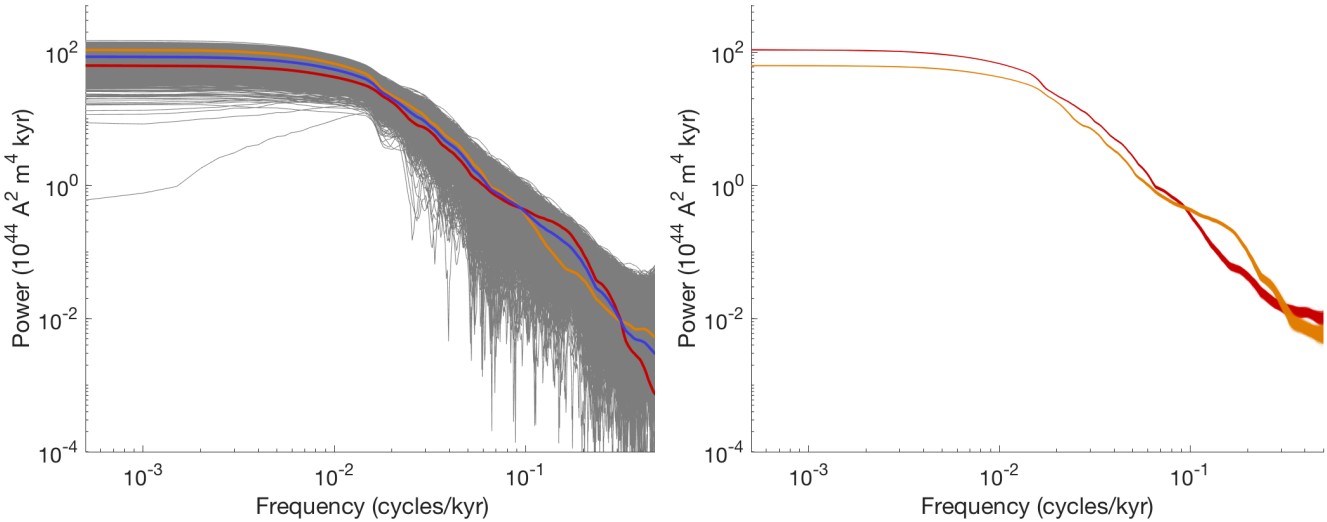

**Figure 4.** Left: low frequency data and error model due to shortness of record. Orange: PSD of Sint-2000. Red: PSD of PADM2M. Blue: mean of PSDs of Sint-2000 and PADM2M ($y_{lf}$). Grey: $5 \cdot 10^3$ samples of the error model $\varepsilon_{lf}$ added to $y_{lf}$. Right: error model based on errors in Sint-2000. Orange: $10^3$ samples of the PSDs computed from "perturbed" Sint-2000 VADMs. Red: $10^3$ samples of the PSDs computed from "perturbed" PADM2M VADMs.

### 4.2.3 High frequencies

We now consider the high frequency behavior of the model and use the CALS10k.2 data. We focus on frequencies between $0.9 - 9.9$ cycles/kyr, where the upper limit is set by the resolution of the CALS10k.2 data. The lower limit is chosen to avoid overlap between the PSDs of CALS10k.2 and Sint-2000/PADM2M. Our choice also acknowledges that the high-frequency part of the PSD for Sint-2000/PADM2M may be less reliable that the PSD of CALS10k.2 for these frequencies. As above, we construct the likelihood $p_{l,\mathrm{hf}}(y|\theta)$ from an equation similar to (23):

$$y_{\mathrm{hf}} = h_{\mathrm{hf}}(\theta) + \varepsilon_{\mathrm{hf}}, \tag{25}$$

where $y_{\mathrm{hf}}$ is the PSD of CALS10k.2 in the frequency range we consider, $h_{\mathrm{hf}}(\theta)$ is a function that maps model parameters to the data and where $\varepsilon_{\mathrm{hf}}$ is the error model.

We base $h_{\mathrm{hf}}(\theta)$ on the PSD of the linear model (see equation (10)) and set

$$h_{\mathrm{hf}}(\theta) = \frac{2D}{\gamma^2 + 4\pi^2 f^2} \cdot \frac{a^2}{a^2 + 4\pi^2 f^2}, \tag{26}$$

where $f$ is the frequency in the range we consider here. Recall that $a^{-1}$ defines the correlation time of the noise in the kyr model.

The error model $\varepsilon_{\mathrm{hf}}$ is Gaussian with mean zero and the covariance is designed to represent errors due to the shortness of the record. This is done, as above, by using 10 kyr simulations of the nonlinear model (6)-(7) with nominal parameter values. We

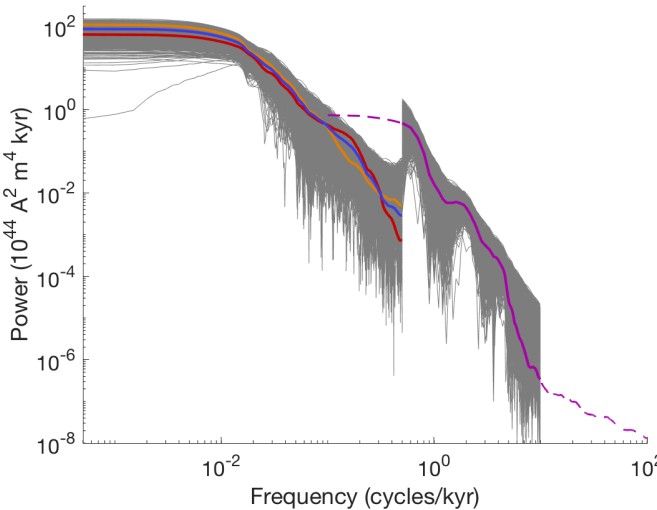

**Figure 5.** Data and error models for low and high frequencies. Orange: PSD of Sint-2000. Red: PSD of PADM2M. Blue: mean of PSDs of Sint-2000 and PADM2M ($y_{\text{lf}}$). Grey (low frequencies): $5 \cdot 10^3$ samples of the error model $\varepsilon_{\text{lf}}$ added to $y_{\text{lf}}$. Dashed purple: PSD of CALS10k.2. Solid purple: PSD of CALS10k.2 at frequencies we consider ($y_{\text{hf}}$). Grey (high frequencies): $5 \cdot 10^3$ samples of the error model $\varepsilon_{\text{hf}}$ added to $y_{\text{hf}}$.

perform 5000 simulations and for each one compute the PSD over the frequency range we consider $(0.9 - 9.9$ cycles/kyr). The covariance matrix computed from these PSDs defines the error model $\varepsilon_{\text{hf}}$, which is illustrated along with the low frequency error model and the data in Figure 5.

This concludes the construction of the likelihood and, together with the prior (see Section 4.1) we have now formulated the
Bayesian formulation of this problem in terms of the posterior distribution (14).

### 4.3 Numerical solution by MCMC

We solve the Bayesian parameter estimation problem numerically by Markov Chain Monte Carlo (MCMC). This means that we use a "MCMC sampler" that generates samples from the posterior distribution in the sense that averages computed over the samples are equal to expected values computed over the posterior distribution in the limit of infinitely many samples. A
(Metropolis-Hastings) MCMC sampler works as follows: the sampler proposes a sample by drawing from a proposal distribution and the sample is accepted with a probability to ensure that the stationary distribution of the Markov chain is the targeted posterior distribution.

We use the affine invariant ensemble sampler, called MCMC Hammer, of Goodman and Weare (2010), implemented in Matlab by Grinsted (2018). The MCMC Hammer is a general purpose ensemble sampler that is particularly effective if there
are strong correlations among the various parameters. The Matlab implementation of the method is easy to use, and requires that we provide the sampler with functions that evaluate the prior distribution and the likelihood, as described above.

In addition, the sampler requires that we define an initial ensemble of ten walkers (two per parameter). This is done as follows. We draw the initial ensemble from a Gaussian whose mean is given by the nominal parameters in Table 3, and whose covariance matrix is a diagonal matrix whose diagonal elements are 50% of the nominal values. The Gaussian is constrained by the upper and lower bounds in Table 3. The precise choice of the initial ensemble, however, is not so important as the ensemble generated by the MCMC hammer quickly spreads out to search the parameter space.

We assess the numerical results by computing integrated auto correlation time (IACT) using the definitions and methods described by Wolff (2004). The IACT is a measure of how effective the sampler is. We generate an overall number of $10^6$ samples, but the number of "effective" samples is $10^6/\text{IACT}$. For all MCMC runs we perform (see Sections 5 and 6), the IACT of the Markov chain is about 100. We discard the first $10 \cdot \text{IACT}$ samples as "burn in", further reducing the impact of the distribution of the initial ensemble. We also ran shorter chains with $10^5$ samples and obtained similar results, indicating that the chains of length $10^6$ are well resolved.

Recall that all MCMC samplers yield the posterior distribution as their stationary distribution, but the specific choice of MCMC sampler defines "how fast" one approaches the stationary distribution and how effective the sampling is (Burn-in time and IACT). In view of the fact that likelihood evaluations are, by our design, computationally inexpensive, we may run (any) MCMC sampler to generate a long chain ($10^6$ samples). Thus, the precise choice of MCMC sampler is not so important for our purposes. We found that the MCMC Hammer solves the problem with sufficient efficiency for our purposes.

The code we wrote is available on github: https://github.com/mattimorzfeld/. It can be used to generate 100,000 samples in a few hours and $10^6$ samples in less than a day. For this reason, we can run the code in several configurations and with likelihoods that are missing some of the factors that comprise the overall feature-based likelihood (15). This allows us to study the impact of each individual data set has on the parameter estimates and it also allows us to assess the validity of some of our modeling choices, in particular with respect to error variances which are notoriously difficult to come by (see Section 6).

## 5  Results

We run the MCMC sampler to generate $10^6$ samples, approximately distributed according to the posterior distribution. We illustrate the posterior distribution by a corner plot in Figure 6. The corner plot shows all 1- and 2-dimensional histograms of the posterior samples. We observe that the four 1-dimensional histograms are well-defined "bumps" whose width is considerably smaller than the assumed parameter bounds (see Table 3) which define the "uninformative", uniform prior. Thus, the posterior probability, which synthesizes the information from the data via the definition of the features, is concentrated over a smaller subset of parameters than the prior probability. In this way, the Bayesian parameter estimation has sharpened the knowledge about the parameters by incorporating the data.

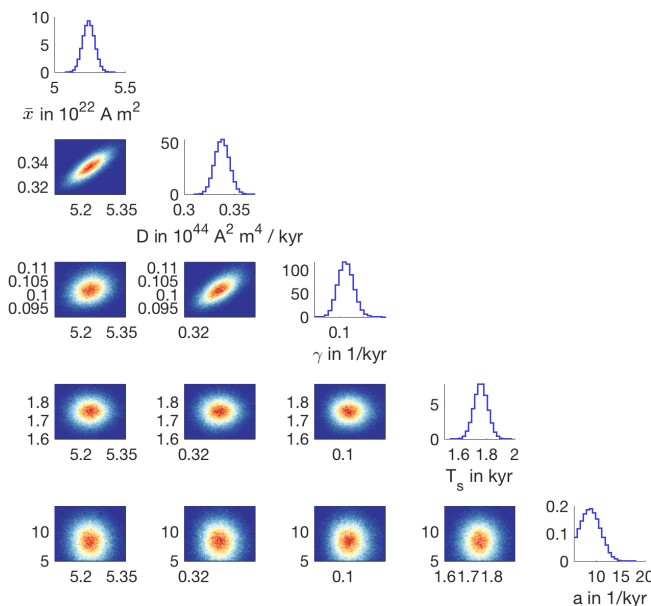

**Figure 6.** 1- and 2-dimensional histograms of the posterior distribution.

| $\bar{x}$ | $D$ | $\gamma$ | $T_s$ | $a$ | $\sigma$ | Rev. rate |
|---|---|---|---|---|---|---|
| in $10^{22}$ Am$^2$ | in $10^{44}$ A$^2$m$^4$ kyr$^{-1}$ | in kyr$^{-1}$ | in kyr | in kyr$^{-1}$ | in $10^{22}$ Am$^2$ | in reversals/Myr |
| 5.23 (0.043) | 0.34 (0.0072) | 0.10 (0.0033) | 1.75 (0.050) | 8.56 (1.93) | 1.77 (0.024) | 4.06 (0.049) |

**Table 4.** Posterior mean and standard deviation (in brackets) of the model parameters and corresponding estimates of reversal rate and VADM standard deviation

The 2-dimensional histograms indicate correlations among the parameters $\theta = (\bar{x}, D, \gamma, T_s, a)^T$, with strong correlations between $\bar{x}$, $D$ and $\gamma$. These correlations can also be described by the correlation coefficients

|  | $\bar{x}$ | $D$ | $\gamma$ | $T_s$ | $a$ |
|---|---|---|---|---|---|
| $\bar{x}$ | 1.00 | 0.78 | 0.20 | 0.02 | $-0.03$ |
| $D$ | 0.78 | 1.00 | 0.64 | 0.02 | $-0.03$ |
| $\gamma$ | 0.20 | 0.64 | 1.00 | $-0.01$ | $-0.02$ |
| $T_s$ | 0.02 | 0.02 | $-0.01$ | 1.00 | 0.00 |
| $a$ | $-0.03$ | $-0.03$ | $-0.02$ | 0.00 | 1.00 |

$$(27)$$

The strong correlation between $\bar{x}$ and $D$ and $\gamma$, is due to the contribution of the reversal rate to the overall likelihood (see
5   Equation (20)) and the dependence of the spectral data on $D$ and $\gamma$ (see Equations (9) and (10)). From the samples, we can also compute means and standard deviations of all five parameters and we show these values in Table 4.

The table also shows the reversal rate and VADM standard deviation that we compute from 2,000 samples of the posterior distribution followed by evaluation of Equations (20) and (13) for each sample. We note that the reversal rate (4.06 reversals/Myr) is lower than the reversal rate we used in the likelihood (4.23 reversals/Myr). Since the posterior standard deviation is 0.049 reversals/Myr, the reversal rate data are about four standard deviations away from the mean we compute. Similarly, the posterior VADM standard deviation (mean value of $1.77 \cdot 10^{22}$ Am$^2$) is also far, as measured by the posterior standard deviation, from the value we use as data ($1.66 10^{22}$ Am$^2$). These large deviations indicate an inconsistency between the VADM standard deviation and the reversal rate. A higher reversal rate could be achieved with a higher VADM standard deviation. The reason is that the reversal rate in Equation (20) can be re-written as

$$r \approx \frac{\gamma}{2\pi} \cdot \exp\left(-\frac{\bar{x}^2}{6\sigma^2}\right) \cdot 10^3 \text{ Myr}^{-1}, \tag{28}$$

using $\sigma \approx \sqrt{D/\gamma}$, i.e., neglecting the correction factor due to sedimentation, which has only a minor effect. Using a time average of $\bar{x} = 5.23 \cdot 10^{22}$ Am$^2$, a reversal rate $r = 4.2$ reversals / Myr, setting $\gamma = 0.1\,\text{kyr}^{-1}$ (posterior mean value), and solving for the VADM standard deviation results in $\sigma \approx 1.86 \cdot 10^{22}$ Am$^2$, which is not compatible with the SINT-2000 and PADM2M data sets (where $\sigma \approx 1.66 \cdot 10^{22}$ Am$^2$). One possible source of discrepancy is that the low-frequency data sets underestimate the standard deviation and also the time average. For example, Ziegler et al. (2008) report a time average VADM of $7.64 \cdot 10^{22}$ Am$^2$ and a standard deviation $\sigma = 2.72 \cdot 10^{22}$ Am$^2$ for paleointensity measurements from the past 0.55 Myr. These measurements are unable to provide any constraint on the temporal evolution of the VADM (in contrast to the SINT-2000 and PADM2M models). Instead, these measurements represent a sampling of the steady-state probability distribution for the dipole moment. The results thus suggest that a larger mean and standard deviation are permitted by paleointensity observations. Using the larger values for the time average and VADM standard deviation, but keeping $\gamma = 0.1\,\text{kyr}^{-1}$ (posterior mean value), leads to a reversal rate of $r \approx 4.27$ reversals per Myr, which is compatible with the reversal rates based on the past 30 Myr in Table 2. It is, however, also possible that the model for the reversal rate has shortcomings. Identifying these shortcomings is a first step in making model improvements and the Bayesian parameter estimation framework we describe is a mathematically and computationally sound tool for discovering such inconsistencies.

The model fit to the spectral data is illustrated in the left panel of Figure 7. Here, we plot 100 PSDs, computed from 2 Myr and 10 kyr model runs, and where each model run uses a parameter set drawn at random from the posterior distribution. For comparison, the figure also shows the PADM2M, Sint-2000 and CALS10k.2 data as well as $5 \cdot 10^3$ realizations of the low- and high-frequency error models. We note that the overall uncertainty is reduced by the Bayesian parameter estimation. The reduction in uncertainty is apparent from the expected errors generating a "wider" cloud of PSDs (in grey) than the posterior estimates (in blue and turquoise). We further note that the PSDs of the models, with parameters drawn from the posterior distribution, fall largely within the expected errors (illustrated in grey). In particular the high-frequency PSDs (the CALS10k.2 range) are well within the errors we imposed by the likelihoods. The low frequencies of Sint-2000/PADM2M are also within the expected errors and so are the high-frequencies beyond the second roll-off due to the sedimentation effects. At intermediate frequencies, some of the PDSs of the model are outside of the expected errors. This indicates a model inconsistency because it

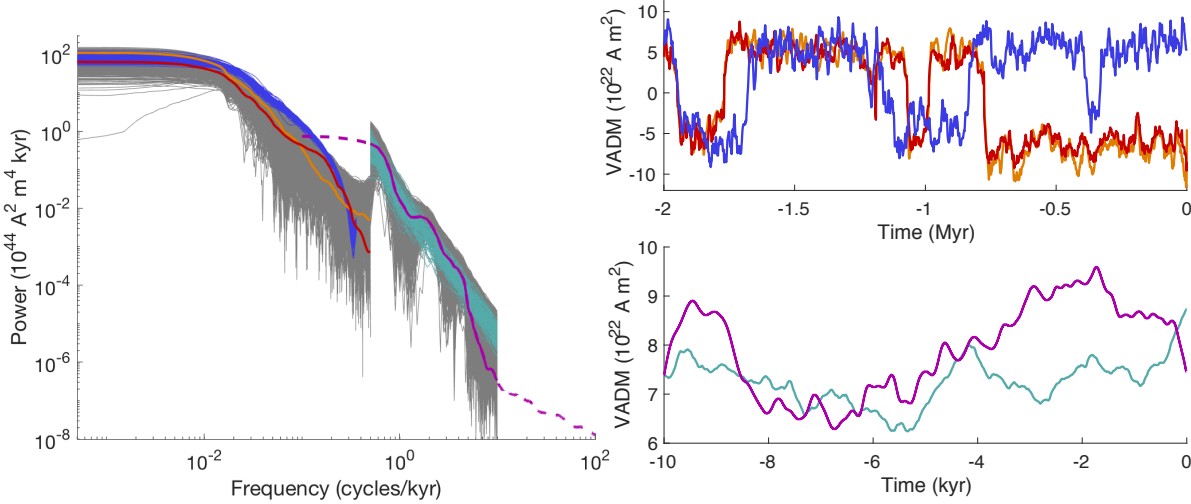

**Figure 7.** Parameter estimation results. Left: PSDs of data and model. Orange: PSD of Sint-2000. Red: PSD of PADM2M. Dashed purple: PSD of CALS10k.2. Solid purple: PSD of CALS10k.2 at frequencies we consider ($y_{hf}$). Grey (low frequencies): $5 \cdot 10^3$ samples of the error model $\varepsilon_{lf}$ added to $y_{lf}$. Grey (high frequencies): $5 \cdot 10^3$ samples of the error model $\varepsilon_{hf}$ added to $y_{hf}$. Dark blue: PSDs of 100 posterior samples of Myr model (with smoothing). Turquoise: PSDs of 100 posterior samples of kyr model with uncorrelated noise. Right, top: Sint-2000 (orange), PADM2M (red) and a realization of the Myr model with smoothing and with posterior mean parameters (blue). Right, bottom: CALS10k.2 (purple) and a realization of the kyr model with correlated noise and with posterior mean parameters kyr model (turquoise).

is difficult to account for the intermediate frequencies with model parameters that fit the other data (spectral and time domain) within the assumed error models. We investigate this issue further in Section 6.

The right panels of Figure 7 show a Myr model run (top) and kyr model run (bottom) using the posterior mean values for the parameters. We note that the model with posterior mean parameter values exhibits qualitatively similar characteristics as
the Sint-2000, PADM2M and CALS10k.2 data. The figure thus illustrates that the feature-based Bayesian parameter estimation, which is based solely on PSD, reversal rates, time average VADM and VADM standard deviation, translates into model parameters that also appear reasonable when a single simulation in the time domain is considered.

In summary, we conclude that the likelihoods we constructed and the assumptions about errors we made lead to a posterior distribution that constrains the model parameters tightly (as compared to the uniform prior). The posterior distribution describes
a set of model parameters that yield model outputs that are comparable with the data in the feature-based sense. The estimates of the uncertainty in the parameters, e.g., posterior standard deviations, however, should be used with the understanding that error variances are not easy to define. For the spectral data, we constructed error models that reflect uncertainty induced by the shortness of the paleomagnetic record. For the time domain data (reversal rate, time average VADM and VADM standard deviation) we used error variances that are smaller than intuitive error variances to account for the fact that the number of
spectral data points (hundreds) is much larger than the number of time domain data points (three data points). Moreover, the

| Configuration | (a) | (b) | (c) | (d) | (e) | (f) |
|---|---|---|---|---|---|---|
| PADM2M & Sint-2000 | ✓ | ✓ | ✓ | ✓ | ✗ | ✓ |
| CALS10k.2 | ✓ | ✗ | ✓ | ✓ | ✗ | ✗ |
| Rev. Rate, time avg., std. dev. | ✓ | ✓ | ✓ | ✗ | ✓ | ✗ |
| $\sigma_{\mathrm{rr}}$ in reversals/Myr | 0.05 | 0.05 | 0.5 | N/A | 0.05 | N/A |
| $\sigma_{\bar{x}}$ in $10^{22}$ Am$^2$ | 0.048 | 0.048 | 0.48 | N/A | 0.048 | N/A |
| $\sigma_{\sigma}$ in $10^{22}$ Am$^2$ | 0.036 | 0.036 | 0.36 | N/A | 0.036 | N/A |

**Table 5.** Configurations for several Bayesian problem formulations. A checkmark means that the data set is used; a cross means it is not used in the overall likelihood construction. The standard deviations ($\sigma$) define the Gaussian error models for the reversal rate, time average VADM and VADM standard deviation.

reversal rate and VADM standard deviation data are far (as measured by posterior standard deviations) from the reversal rate and VADM standard deviation of the model with posterior parameters. As indicated above, this discrepancy could be due to inconsistencies between spectral data and time domain data, which we will study in more detail in the next section.

## 6 Discussion

We study the effects the independent data sets have on the parameter estimates and also study the effects of different choices for error variances for the time domain data (reversal rate, time average VADM and VADM standard deviation). We do so by running the MCMC code in several configurations. Each configuration corresponds to a posterior distribution and, therefore, to a set of parameter estimates. The configurations we consider are summarized in Table 4 and the corresponding parameter estimates are reported in Table 5. Configuration (a) is the default configuration described in the previous sections. We now discuss the other configurations in relation to (a) and in relation to each other.

Configuration (b) differs from configuration (a) in that the CALS10k.2 data are not used, i.e., we do not include the high-frequency component, $p_{l,\mathrm{hf}}(y|\theta)$ in the feature-based likelihood (15). Configurations (a) and (b) lead to nearly identical posterior distributions and, hence, nearly identical parameter estimates with the exception of the parameter $a$, which controls the correlation of the noise on the kyr time scale. The differences and similarities are apparent when we compare the corner plots of the posterior distributions of configurations (a), shown in Figure 6, and of configuration (b), shown in Figure 8. The corner plots are nearly identical except for the bottom row of plots which illustrates marginals of the posterior related to $a$. We note that the posterior distribution over $a$ is nearly identical to its prior distribution. Thus, the parameter $a$ is not constrained by the data used in configuration (b), which is perhaps not surprising because $a$ only appears in the Bayesian parameter estimation problem via the high-frequency likelihood $p_{l,\mathrm{hf}}(y|\theta)$. Moreover, since $p_{l,\mathrm{lf}}(y|\theta)$ and $p_{l,\mathrm{td}}(y|\theta)$ are independent of $a$, the marginal of the posterior distribution of configuration (b) over the parameter $a$ is independent of the data. More interestingly, however, we find that all other model parameters are estimated to have nearly the same values, independently of whether CALS10k.2 being used during parameter estimation or not. This latter observation indicates that the model is self-consistent and consistent with the

| Configuration | (a) | (b) | (c) |
|---|---|---|---|
| $\bar{x}$ in $10^{22}$ $Am^2$ | 5.23 (0.043) | 5.23 (0.042) | 3.56 (0.26) |
| $D$ in $10^{44}$ $A^2m^4$ $kyr^{-1}$ | 0.34 (0.0072) | 0.34 (0.0072) | 0.13 (0.014) |
| $\gamma$ in $kyr^{-1}$ | 0.10 (0.0033) | 0.10 (0.0033) | 0.081 (0.0052) |
| $T_s$ in kyr | 1.75 (0.050) | 1.74 (0.050) | 1.68 (0.14) |
| $a$ in $kyr^{-1}$ | 8.56 (1.93) | 22.45 (10.01) | 11.69 (3.91) |
| $\sigma$ in $10^{22}$ $Am^2$ | 1.77 (0.024) | 1.77 (0.023) | 1.22 (0.060) |
| Rev. rate in reversals/Myr | 4.06 (0.049) | 4.06 (0.023) | 3.34 (0.52) |

| Configuration | (d) | (e) | (f) |
|---|---|---|---|
| $\bar{x}$ in $10^{22}$ $Am^2$ | 5.04 (2.91) | 5.56 (0.048) | 5.04 (2.88) |
| $D$ in $10^{44}$ $A^2m^4$ $kyr^{-1}$ | 0.094 (0.015) | 0.44 (0.021) | 0.093 (0.015) |
| $\gamma$ in $kyr^{-1}$ | 0.078 (0.0063) | 0.14 (0.014) | 0.077 (0.0064) |
| $T_s$ in kyr | 1.64 (0.19) | 2.98 (1.15) | 1.64 (0.19) |
| $a$ in $kyr^{-1}$ | 12.92 (4.79) | 22.59 (10.08) | 22.37 (10.13) |
| $\sigma$ in $10^{22}$ $Am^2$ | 1.08 (0.074) | 1.66 (0.036) | 1.07 (0.075) |
| Rev. rate in reversals/Myr | 2.89 (4.11) | 4.23 (0.050) | 2.83 (4.10) |

**Table 6.** Posterior parameter estimates (mean and standard deviation, in brackets) and corresponding VADM standard deviation ($\sigma$) and reversal rates for five different set ups (see Table 5).

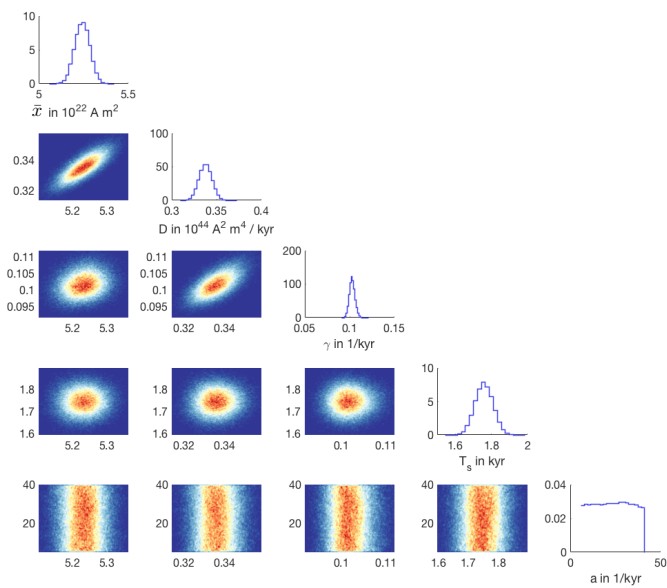

**Figure 8.** 1- and 2-dimensional histograms of the posterior distribution of configuration (b).

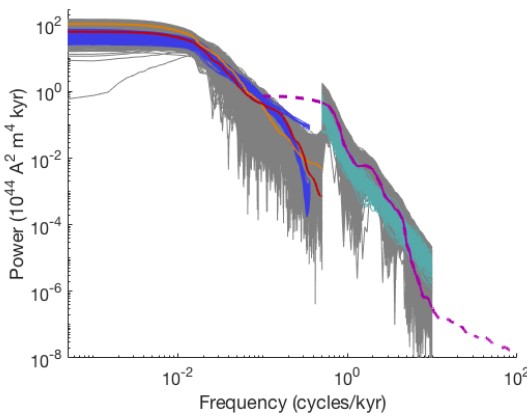

**Figure 9.** PSDs of data and model with parameters drawn from the posterior distribution of configuration (c). Orange: PSD of Sint-2000. Red: PSD of PADM2M. Dashed purple: PSD of CALS10k.2. Solid purple: PSD of CALS10k.2 at frequencies we consider ($y_{\text{hf}}$). Grey (low frequencies): $5 \cdot 10^3$ samples of the error model $\varepsilon_{\text{lf}}$ added to $y_{\text{lf}}$. Grey (high frequencies): $5 \cdot 10^3$ samples of the error model $\varepsilon_{\text{hf}}$ added to $y_{\text{hf}}$. Dark blue: PSDs of 100 posterior samples of Myr model (with smoothing). Turquoise: PSDs of 100 posterior samples of kyr model with uncorrelated noise.

data on the Myr and kyr time scales; in the context of our simple stochastic model the data from CALS10k.2 mostly constraints the noise correlation parameter $a$.

Configuration (c) differs from configuration (a) in the error variances for the time domain data (reversal rate, time average VADM and VADM standard deviation). With the larger values used in configuration (c), the spectral data are emphasized during

the Bayesian estimation which also leads to an overall better fit of the spectra. This is illustrated in Figure 9, where we plot the 100 PSDs generated by 100 (independent) simulations with the model with parameters drawn from the posterior distribution of configuration (c). For comparison, we also plot the PSDs of PADM2M, Sint-2000, CALS10k.2 and $5 \cdot 10^3$ realizations of the high- and low-frequency error model. In contrast to configuration (a) (see Figure 7), we find that the PSDs of the model of configuration (c) are all well within the expected errors. One the other hand, the reversal rate drops to about 3 reversals/Myr,

and the time average VADM and VADM standard deviation also decrease significantly as compared to configuration (a). This is caused by the posterior mean of $D$ being decreased by more than 50%, while $\gamma$ and $T_s$ are comparable for configurations (a)-(c). The fact that the improved fit of the PDSs comes at the cost of a poor fit of the reversal rate, time average and standard deviation is another indication of an inconsistency between the reversal rate and the VADM data sets. As indicated above, one of the strengths of the Bayesian parameter estimation framework we describe here is to be able to identify such inconsistencies.

Once identified, one can try to fix the model. For example, we can envision a modification of the functional form of the drift term in (2). A nearly linear dependence of the drift term on $x$ near $x = \bar{x}$ is supported by the VADM data sets, but the behavior near $x = 0$ is largely unconstrained. Symmetry of the underlying governing equations suggests that the drift term should vanish and the functional form adopted in (2) is just one way that a linear trend can be extrapolated to $x = 0$. Other functional forms that lower the barrier between the potential wells would have the effect of increasing the reversal rate. This simple change to

the model could bring the reversal rate into better agreement with the time average and standard deviation of the VADM data sets.

In configuration (d), the spectral data are used, but the time domain data are not used (which corresponds to infinite $\sigma_{rr}$, $\sigma_\sigma$ and $\sigma_{\bar{x}}$). We note that the posterior means and variances of all parameters are comparable for configurations (c), where the error variances of the time domain data are "large", and (d), where the error variance of the time domain data are "infinite". Thus, the impact of the time domain data is minimal if the error variances of the time domain data are large. The reason is that the number of spectral data points is larger (hundreds) than the number of time domain data (three data points: reversal rate, time average VADM and VADM standard deviation). When the error variances of the time domain data decrease, the impact these data have on the parameter estimates increases. We further note that the parameter estimates of configurations (c) or (d) are quite different from the parameter estimates of configuration (a) (see above). For an overall good fit of the model to the spectral and time domain data, the error variances for the time domain data must by small, as in configuration (a). Otherwise, the reversal rates are too low. Small error variances, however, imply (relatively) large deviations between the time domain data and the model predictions. Small error variances also comes at the cost of not necessarily realistic posterior variances.

Comparing configurations (d) and (e), we note that if only the spectral data are used, the reversal rates are unrealistically small (nominally 1 reversal per Myr). Moreover, the parameter estimates based on the spectral data are quite different from the estimates we obtain when we use the time domain data (reversal rate, time average VADM and VADM standard deviation). This is further evidence that either the model has some inconsistencies, or that the reversal rate and the VADM standard deviation are not consistent. Specifically, our experiments suggest that a good match to spectral data requires a set of model parameters that is quite different from the set of model parameters that lead to a good fit to the reversal rate, time average VADM and VADM standard deviation. Experimenting with different functional forms for the drift term is one strategy for achieving better agreement between the reversal rate, the time average VADM and VADM standard deviation.

Comparing configurations (d) and (f), we can further study the effects that the CALS10k.2 data have on parameter estimates (similarly to how we compared configurations (a) and (b) above). The results, shown in Table 6, indicate that the parameter estimates based on configurations (d) and (f) are nearly identical, except in the parameter $a$ that controls the time correlation of the noise on the kyr time scale. This confirms what we already found by comparing configurations (a) and (b): the CALS10k.2 data are mostly useful for constraining $a$. These results, along with configurations (a) and (b), suggest that the model is self consistent with the independent data on the Myr scale (Sint-2000 and PADM2M) and on the kyr scale (CALS10k.2). Our experiments, however, also suggest that the model has difficulties to reconcile the spectral and time domain data.

Finally, note that the data used in configuration (d) does not inform the parameter $\bar{x}$, and configuration (f) does not inform $\bar{x}$ or $a$. If the data do not inform the parameters, then the posterior distribution over these parameters is essentially equal to the prior distribution, which is uniform. This is illustrated in Figure 10, where we show the corner plot of the posterior distribution of configuration (f). We can clearly identify the uniform prior in the marginals over the parameters $\bar{x}$ and $a$. This means that the Sint-2000 and PADM2M data only constrain the parameters $D$, $\gamma$ and $T_s$.

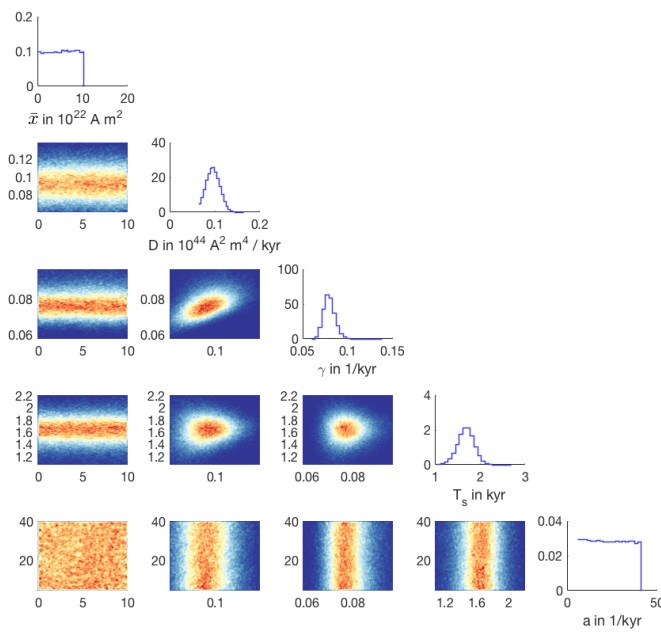

**Figure 10.** 1- and 2-dimensional histograms of the posterior distribution of configuration (f).

## 7 Examples of applications of the model

The Bayesian estimation technique we describe leads to a model with stochastic parameters whose distributions are informed by the paleomagnetic data. Moreover, we ran a large number of numerical experiments to understand the limitations of the model, to discover inconsistencies between the model and the data and to check our assumptions about error modeling. This

process results in a well-understood and well-founded stochastic model for selected aspects of the long term behavior of the geomagnetic dipole field. We believe that such a model can be useful for a variety of purposes, including testing hypotheses about selected long-term aspects of the geomagnetic dipole.

For example, it was noted by Ziegler et al. (2011) that the VADM (time averaged) amplitude during the past chron was slightly lower than during the previous chron. Specifically, the time averaged VADM for $-0.78 < t < 0$ Myr is $E(x) = 6.2 \cdot$

$10^{22}$ Am$^2$ but for $-2 < t < -0.78$ Myr the time average is $E(x) = 4.8 \cdot 10^{22}$ Am$^2$. A natural question is: is this increase in the time average significant or is it due due to random variability? We investigate this question using the model whose parameters are the posterior mean values of configuration (a) (the configuration that leads to an overall good fit to all data). Specifically, we perform 10,000 simulations of duration $0.78$ Myr and 10,000 independent simulations of duration $1.22$ Myr. For each simulation, we compute the time average, which allows us to estimate the standard deviation of the difference in means

(assuming no correlation between the two time intervals). We found that this standard deviation is about $0.46 \cdot 10^{22}$ Am$^2$, which is much smaller than the differences in VADM time averages of $1.4 \cdot 10^{22}$ Am$^2$. This suggests that the increase in time averaged VADM is likely not due to random variability.

A similar approach can be applied to the question of changes in the reversal rate over geological time. The observed reversal rate over the past 30 Myr is approximately $4.26 \, \text{Myr}^{-1}$. When the record is divided into 10-Myr intervals, the reversal rate varies about the average, with a standard deviation of about $0.49 \, \text{Myr}^{-1}$ (see Table 2). These variations are within the expected fluctuations for the stochastic model. Specifically, we can use an ensemble of $10^5$ simulations, each of duration 10 Myr to

compute the average and standard deviation of the reversal rate The results, obtained by using nominal parameters and posterior mean parameters of configurations (a) (overall good fit to all data) and (e) (emphasis on reversal rate data) are shown in Table 7. As already indicated in Section 4.2.1, the standard deviation from the geomagnetic polarity time scale is comparable to the

| | Nominal parameter values | Configuration (a) | Configuration (e) |
|---|---|---|---|
| Average reversal rate (in $\text{Myr}^{-1}$) | 4.50 | 3.76 | 3.56 |
| Standard deviation (in $\text{Myr}^{-1}$) | 0.68 | 0.63 | 0.59 |

**Table 7.** Average reversal rate and and standard deviation of an ensemble of $10^5$ simulations of duration 10 Myr, filtered to a resolution of 30 kyr. The simulations are done with nominal parameter values (Table 3), or the posterior mean values of configurations (a) and (e).

standard deviation we compute via the model. The observed reversal rate for the 10-Myr interval between 30 and 40 Myr, however, is approximately $2.0 \, \text{Myr}^{-1}$, which departs from the $0 - 30$ Myr average by more than three standard deviations. This

suggests that the reversal rate between 30 and 40 Ma cannot be explained by natural variability in the model. Instead, it suggests that model parameters were different before 30 Ma, implying that there was a change in the operation of the geodynamo.

## 8   Summary and conclusions

We consider parameter estimation for a model of Earth's axial magnetic dipole field. The idea is to estimate the model parameters using data that describe Earth's dipole field over kyr and Myr time scales. The resulting model, with calibrated parameters,

is thus a representation of Earth's dipole field on these time scales. We formulated a Bayesian estimation problem in terms of "features" that we derived from the model and data. The data include two time series (Sint-2000 and PADM2M) that describe the strengths of Earth's dipole over the past 2 Myr, a shorter record (CALS10k.2) that describes dipole strength over the past 10 kyr, as well as reversal rates derived from the geomagnetic polarity time scale. The features are used to synthesize information from these data sources (that had previously been treated separately).

Formulating the Bayesian estimation problem requires defining anticipated model error. We found that the main source of uncertainty is the shortness of the paleomagnetic record and constructed error models to incorporate this uncertainty. Numerical solution of the feature-based estimation problem is done via conventional Markov chain Monte Carlo (an affine invariant ensemble sampler). With suitable error models, our numerical results indicate that the paleomagnetic data constrain all model parameters in the sense that the posterior probability mass is concentrated on a smaller subset of parameters than the prior

probability. Moreover, the posterior parameter values yield model outputs that fit the data in a precise, feature-based sense, which also translates into a good fit by other, more intuitive measures.

A main advantage of our approach (Bayesian estimation with an MCMC solution) is that it allows us to understand the limitations and remaining (posterior) uncertainties of the model. After parameter estimation, we thus have produced a reliable, stochastic model for selected aspects of the long term behavior of the geomagnetic dipole field whose limitations and errors are well-understood. We believe that such a model is useful for hypothesis testing and have given several examples of how the model can be used in this context. Another important aspect of our overall approach is that it can reveal inconsistencies between model and data. For example, we ran a suite of numerical experiments to assess the internal consistency of the data and the underlying model. We found that the model is self-consistent on the Myr and kyr time scales, but we discovered inconsistencies that make it difficult to achieve a good fit to all data simultaneously. It is also possible that the data themselves are not entirely self consistent in this regard. Our methodology does not resolve these questions, but once inconsistencies are identified, several strategies can be pursued to resolve them, e.g., improving the model or resolving consistency issues of the data themselves. Our conceptual and numerical framework can also be used to reveal the impact that some of the individual data sets have on parameter estimates and associated posterior uncertainties. In this paper, however, we focused on describing the mathematical and numerical framework and only briefly mention some of the implications.

*Code and data availability.* The code and data used in this paper is available on github: https://github.com/mattimorzfeld

*Competing interests.* No competing interests are present.

*Acknowledgements.* We thank Dr. Johannes Wicht of Max Planck Institute for Solar System Research, and an anonymous reviewer for careful comments that helped improving our paper. We also thank Dr. Andrew Tangborn of NASA GSFC for interesting discussion and useful comments. MM gratefully acknowledges support by the National Science Foundation under grant DMS-1619630 and by the Alfred P. Sloan Foundation. B.B. acknowledges support from grant EAR-164464 from the National Science Foundation.

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
