# Peer review of "A comprehensive model for the kyr and Myr time scales of Earth's axial magnetic dipole field"

_Nonlinear Processes in Geophysics, 2018_

## Referee Comment (RC1) · Johannes Wicht (Referee) · 7 Feb 2019

The geomagnetic field varies on time scales from about a year to several tens of million years. The vastly different time scales have been revealed by various data sources, all having their specific inherent problems and limitations. In their pioneering work from 2001, Hoyng and Schmitt suggest that at least the axial dipole variations on time scales from millennia to some millions of years can be describe by the Langevin equation, a simple stochastic differential equation used, for example, to model Brownian motion. In recent years, the respective model has been refined to include the effects of correlated noise, random errors, or the limited time resolution of sedimentary data. This

manuscript introduces another refinement, a Bayesian approach that allows to incorporate different types of 'data' in a probabilistic manor to constrain the (five) parameters of the stochastic model. The paper is interesting and well written, but requires a few additional clarifications here and there (see below). The authors already mention that their analysis reveals difficulties and/or inconsistencies, but the paper remains too vague at this point. There problem could result from the different (and inconsistent) treatment of time-domain and frequency domain data, but this is hard to judge from the manuscript. A generally more critical discussion of the approach also seems in order. In addition, it remains unclear whether the results reveal anything new about geomagnetic field variations. A few respective additional sentences, for example in the conclusion, would certainly strengthen the paper.

Mayor points:

1) Data: the stochastic model is constraint based on paleomagnetic and archeomagnetic data, which both have many problems. The dating uncertainties and the smoothing due to the lock-in-time in sediments are mentioned in the text, but there are more. While there is little the authors can do about this, some more critical discussion and application seems in order. For example, the fact that the two paleomagnetic models show sizable differences suggests using a large uncertainty when modelling the respective data. The reversal rate is another example. How well can one determine the reversal rate based on a 30 Myr record when the underlying process is Poissonian?

2) Geomagnetic power spectra: The power spectra and the respective models are not discussed in any detail. Also missing is an explanation how the spectra are incorporated in the Bayesian approach. Combining hundreds of 'frequency data points' with only three 'time domain data points' may not be the best way to proceed and certainly required some care. A parametrization or identification of the most important spectral features seems the way to go here. See Baerenzung et al. (2018) for such an alternative approach. Figure 3 and 7 suggest that neither the spectra from the stochastic model runs nor the theoretical spectra do a very convincing job in replicating the spectra from the data, at least when it comes to the location of 'knees' or of typical slopes. Some additional text seems in order here. How much can we trust the spectra and in which frequency range? What are the limitation of the stochastic model when it comes to the spectra? How much agreement can we reasonably expect? Please also remind the reader why the Sint-2000/PADM2M and CALS10K.2 spectra are so different where they overlap in frequencies. Why does figure 7 show a different range than figure 3? Do you really think that the stochastic model can capture the high frequency part shown in figure 7?

3) Critical discussion: The authors already point out some problems or 'inconsistencies' in the sense that their model cannot capture all 'data' convincingly. This should be discussed in more detail. Is this a problem in the data or is the model too simple? The deficiencies in describing the spectra seems to imply the latter. It seems to me that the model is doing OK for describing the long-term variations where any complexity due to the convection and the internal dynamo process may not matter so much. Implementing a archeomagnetic (or even historic model) seem then overambitious. The authors should also discuss whether we can we learn anything about the geodynamo from this approach?

Minor points:

1) Please check the way you cite. You seem to mix up citep and citet.

2) Abstract, last sentence: Bayesian reasoning is frequently used for combining different data. What exactly is new in your approach?

3) Same sentence: . . . data sets, which is particularly . . .

4) Page 1, line 23: What do you mean by: . . . even basic analytical calculations are often intractable?

5) Page 2, line 5 and following: To my knowledge, the basic concept has been introduced by Hoyng, Schmitt and Ossendrijver in a series of papers in 2001 and 2002

(see below). Please give them credit. What exactly are the new ingredient in the B13 model? Note that Meduri & Wicht (2016) claim that the linearization used later in the paper (and which may also be a component of model B13) may only be of limited use.

6) Page 2, line 54: Define SDE.

7) Page 2, line 19: Buffett and Puranam (2017) try to mimic the effects of sedimentation ...

8) Page 3, line 29: ... to reduce the influence of non-dipole components and various error sources ...

9) Page 4, line 3: CALSK10k.2 sampled at an interval of 1 year? Is this really the model's resolution?

10) Page 5, line 14: At least set "Brownian motion" in quotation marks.

11) Page 5, line 18: Concerning a constant D, see Meduri & Wicht (2016).

12) Page 5, line 22: ... well potential potential ...

13) Caption of fig. 2: ... and potential U(x), with U'(x)=- v(x) ... Or use an integral formulation.

14) Page 6, line 4: Could you explain iid in a few words for the non-experts?

15) Page 6, line 9: ... are affected by affected by ...

16) Caption of fig. 3: There is something wrong with the sentences.

17) Page 8: Because an SDE is noisy ... Well, no surprise there. Could you be more specific? Wouldn't this also depend on the noise parameters? How expensive is an SDE integration? How long do you have to integrate?

18) Page 8, line 10: The comparison does not look too good. Please discuss.

19) Page 8, line 14: ... the time averaged value of the absolute value of x(t) ...

20) Page 8, line 23: approximating -> approximate

21) Page 8, line 27: The SD is not very close to the observation. Please discuss.

22) Page 9, line 4: Sampled once per year . . . (see above).

23) Page 9, line 18: Discuss the comparison.

24) Page 9, line 30: . . . the bound may be overly pessimistic . . . Well, the problem is not solved yet, but it looks like electron-electron interaction could at best only have a mild effect. Anyway, there is no need to dive into this topic in the paper and I would simply drop this sentence.

25) Page 10, top: Note that the stochastic model represents longer statistical time scales. Arguing with flow velocities is of limited use here.

26) Sections 4.2.2 and 4.2.3: Please provide more details here. (See comment above)

27) Figure 6: Provide colorbar. Discuss a bit more. How well are the parameters constrained, for example compared to the priors?

28) Page 17, line 2: The "however" seems out of place.

29) Page 17, line 5: . . . the impact of each . . .

30) Page 21, line 1: . . . in the context of our simplistic model CALS10k.2 mostly constraints . . .

31) Page 21, line 30: Every geomagnetic data point is indeed the result of hard work, but why is this a challenge for the model described here?

32) Page 22, top: Incorporating different type of 'data' in a Bayesian approach is a standard application. Please point out the specific novelty in your approach.

33) Page 22, line 4: "We use the full paleomagnetic record"? The years of hard work have resulted in more than just Sint-2000 and PADM2M.

Baerenzung, J., Holschneider, M., Wicht, J. Sanchez, S. and Lesur, V., Modeling and Predicting the Short-Term Evolution of the Geomagnetic Field, J. Geophys Res. (solid Earth), 123, pp.4539-4560, 2018.

Hoyng, P., Ossendrijver, M.A.J.H. and Schmitt, D., The geodynamo as a bistable oscillator,Geosphys. Astrophys. Fluid. Dyn., 94, 2001.

Hoyng, P., Schmitt, D. and Ossendrijver, M.A.J.H., A theoretical analysis of the observed variability of the geomagnetic dipole field, Phys. Earth Planet. Int., 130, pp. 143-157, 2002.

Schmitt, D., Ossendrijver, M.A.J.H. and Hoyng, P., Magnetic field reversals and secular variation in a bistable geodynamo model, Phys. Earth Planet. Int., 125, pp. 119-124, 2001.

---

## Referee Comment (RC2) · Anonymous Referee #2 · 7 Feb 2019

Review

The authors take an interesting Bayesian approach, but there is a number of issues which require an improvement of the manuscript. The developed stochastic model uses of paleomagnetic and archeomagnetic data. How do the authors treat uncertainties in dating, smooth the data and carry out other data massages?

How can one combine many frequency-domain data points with a few time-domain data points?

Please, discuss the geomagnetic power spectra and the respective models in some more detail. How do they incorporate them in the Bayesian approach?

What do we learn from their approach about geomagnetic field variations or about the dynamo process in the Earth using their model?

Interactive
comment

---

## Author Comment (AC1) · 19 Mar 2019

**A comprehensive model for the kyr and Myr time scales of Earth's axial magnetic dipole field – response to Reviewers' comments**

Matthias Morzfeld, Bruce A. Buffett

We thank both Reviewers for their careful reviews and good suggestions. We incorporated most of the Reviewers' comments into our revised manuscript. A detailed, point-by-point response is provided below. We repeat the Reviewers' comments in italic and our responses appear in standard font. We also provide a latex-diff that can be used to track all changes from our original submission to this revised version. The line numbers below (and in the reviews) refer to the original submission which we also provide.

In addition to the changes described below, we revised the entire manuscript based on the Reviewers' comments. This has led us to remove Figure 2 (it was not really a necessary figure). We split Figure 3 into two figures and included additional information about our error models in a revised Figure 7 (all Figure numbers refer to the original submission). We further included several new figures. We revised Sections 3 and 4 for clarity and brevity, and expanded many of our explanations (as requested by the Reviewers), which causes the revised manuscript to be longer than the original submission. We included a new section that touches on the usefulness of our approach. All these changes are inspired by the Reviewers' comments, but not all of them are directly related to the Reviewers' major or minor points.

We further included a correction factor for the approximate standard deviation (see Equation (13) of the revised manuscript). The correction factor is (for most relevant parameter values) near one and, for that reason, including the correction causes only minor changes in the numerical (posterior) estimates and does not change our conclusions or overall results.

**Response to Reviewer 1 (Johannes Wicht)**

*The geomagnetic field varies on time scales from about a year to several tens of million years. The vastly different time scales have been revealed by various data sources, all having their specific inherent problems and limitations. In their pioneering work from 2001, Hoyng and Schmitt suggest that at least the axial dipole variations on time scales from millennia to some millions of years can be describe by the Langevin equation, a simple stochastic differential equation used, for example, to model Brownian motion. In recent years, the respective model has been refined to include the effects of correlated noise, random errors, or the limited time resolution of sedimentary data. This manuscript introduces another refinement, a Bayesian approach that allows to incorporate different types of "data" in a probabilistic manor to constrain the (five) parameters of the stochastic model. The paper is interesting and well written, but requires a few additional clarifications here and there (see below). The authors already mention that their analysis reveals difficulties and/or inconsistencies, but the paper remains too vague at this point. There problem could result from the different (and inconsistent) treatment of time-domain and frequency domain data, but this is hard to judge from the manuscript. A generally more critical discussion of the approach also seems in order. In addition, it remains unclear whether the results reveal anything new about geomagnetic field variations. A few respective additional sentences, for example in the conclusion, would certainly strengthen the paper.*

We thank the Reviewer for the suggestions. We revised the manuscript in several places to tighten our statements about model and data inconsistencies (see also below). We further extended our

discussion and provide more details about the limitations and advantages of our overall approach. We have added a new section to indicate why and how our approach is useful for studying the geomagnetic field. In short, a model is not a reliable scientific tool unless the model parameters are carefully chosen (based on "data") and unless limitations and uncertainties of the model are understood. We describe here a framework for doing just that for a stochastic model of the geomagnetic field.

**Response to major points**

1. *Data: the stochastic model is constraint based on paleomagnetic and archeomagnetic data, which both have many problems. The dating uncertainties and the smoothing due to the lock-in-time in sediments are mentioned in the text, but there are more. While there is little the authors can do about this, some more critical discussion and application seems in order. For example, the fact that the two paleomagnetic models show sizable differences suggests using a large uncertainty when modelling the respective data. The reversal rate is another example. How well can one determine the reversal rate based on a 30 Myr record when the underlying process is Poissonian?*

   The Reviewer is right in that there is a long list of problems. We did our best to bring up the ones that are most severe and most relevant for our purposes, but we do not claim that our list of problems is complete. We do use a large uncertainty when modeling the two paleomagnetic data sets. The uncertainties that arise from differences of the two data sets, however, are small in comparison to uncertainties that arise due to the fact that the time series is "only" 2 Myrs long. We describe this in detail in Section 4.2.2, see in particular Figure 4 (of the new submission). We have added explanations that the reversal rate cannot be accurately derived form a 30 Myr record. As is the case with the paleomagnetic data, the main source of this uncertainty is the "shortness" of the geomagnetic record. We have revised the manuscript throughout to highlight this point. How well one can determine the reversal rate based on a 30 Myr record when the underlying process is Poissonian depends on the rate of the Poissonian process and on how certain one is about one's knowledge of that rate.

2. *Geomagnetic power spectra: The power spectra and the respective models are not discussed in any detail. Also missing is an explanation how the spectra are incorporated in the Bayesian approach. Combining hundreds of "frequency data points" with only three ?time domain data points? may not be the best way to proceed and certainly required some care. A parametrization or identification of the most important spectral features seems the way to go here. See Baerenzung et al. (2018) for such an alternative approach. Figure 3 and 7 suggest that neither the spectra from the stochastic model runs nor the theoretical spectra do a very convincing job in replicating the spectra from the data, at least when it comes to the location of "knees" or of typical slopes. Some additional text seems in order here. How much can we trust the spectra and in which frequency range? What are the limitation of the stochastic model when it comes to the spectra? How much agreement can we reasonably expect? Please also remind the reader why the Sint-2000/PADM2M and CALS10K.2 spectra are so different where they overlap in frequencies. Why does figure 7 show a different range than figure 3? Do you really think that the stochastic model can capture the high frequency part shown in figure 7?*

   The power spectral densities are computed from the time series (Sint-2000, PADM2M, CALS10k.2) using the multi-taper technique of Constable and Johnson (2005). Sections 4.2.2 and 4.2.3 describe how the PSDs are incorporated in the Bayesian approach (via feature-based likelihoods). We do not agree that a parametrization or identification of the most important

spectral features is obviously "the way to go here" as it may not be straightforward to extract these features from the data. We believe both approaches have advantages and disadvantages, but decided to stick to our approach of tuning the error variances. We also did not find Baerenzung et al. (2018) particularly useful for learning about how to parametrize or identify the most important spectral features. Nonetheless, we added this paper to our references. We have added additional text to discuss in more detail the "quality" of the fit. We have also included additional figures to make our points more clear. We have overlooked this before and thank the Reviewer for brining up that the original submission indeed lacked a lot of required explanations. We now use the same frequency range for (what used to be) Figures 3 and 7.

3. *Critical discussion: The authors already point out some problems or "inconsistencies" in the sense that their model cannot capture all "data" convincingly. This should be discussed in more detail. Is this a problem in the data or is the model too simple? The deficiencies in describing the spectra seems to imply the latter. It seems to me that the model is doing OK for describing the long-term variations where any complexity due to the convection and the internal dynamo process may not matter so much. Implementing a archeomagnetic (or even historic model) seem then overambitious. The authors should also discuss whether we can we learn anything about the geodynamo from this approach?*

    We have added explanations and clarifications and discuss the inconsistencies in model and data in more detail. It is difficult to come to a indisputable conclusion as to whether the inconsistencies are (mainly) due to the data or due to the model being too simple. Clearly, the model is simple, but also the data are known to be inconsistent. We explain these difficulties, but ultimately are not able to judge if the model is too simple or the data too inconsistent. We added a new section to discuss how our approach can be useful for studying the geomagnetic field (Section 7). We believe that our main finding about modeling of the geomagnetic field is that uncertainties in the data are dominated by errors that arise from the shortness of the record. Errors that arise due to the specifics of how the data are obtained (e.g., differences between PADM2M and Sint-2000) seem minor in comparison.

**Response to minor points**

1. *Please check the way you cite. You seem to mix up citep and citet.*

    We thank the Reviewer for bringing this to our attention. This was a careless mistake and we fixed it.

2. *Abstract, last sentence: Bayesian reasoning is frequently used for combining different data. What exactly is new in your approach?*

    We revised this sentence and hope that our revision is acceptable for the Reviewer. While it is true that Bayesian methods are used to combine different data, it is not well-understood what to do in a situation when several data describe the same quantity, but not necessarily in a consistent way. It is difficult to explain in a sentence or two why our approach is useful in this situation, which is why we decided not to include a (vague) description in the abstract.

3. *Same sentence: ... data sets, which is particularly ...*

    We fixed this typo.

4. *Page 1, line 23: What do you mean by: ... even basic analytical calculations are often intractable?*

We dropped any mention of analytical calculations.

5. *Page 2, line 5 and following: To my knowledge, the basic concept has been introduced by Hoyng, Schmitt and Ossendrijver in a series of papers in 2001 and 2002 (see below). Please give them credit. What exactly are the new ingredient in the B13 model? Note that Meduri & Wicht (2016) claim that the linearization used later in the paper (and which may also be a component of model B13) may only be of limited use.*

   We included the suggested references and also discussed Meduri & Wicht (2016) in more detail.

6. *Page 2, line 54: Define SDE.*

   We fixed this issue.

7. *Page 2, line 19: Buffett and Puranam (2017) try to mimic the effects of sedimentation*

   We use the suggested formulation in the revised manuscript.

8. *Page 3, line 29: . . . to reduce the influence of non-dipole components and various error sources . . .*

   This sentence disappeared in our revision.

9. *Page 4, line 3: CALSK10k.2 sampled at an interval of 1 year? Is this really the model's resolution?*

   We have added clarification of this issue. CALS10k.2 can be sampled at any rate (suggested is 1 yr to 200 yrs) and the nominal resolution is about 100 yrs.

10. *Page 5, line 14: At least set "Brownian motion" in quotation marks.*

    We do not understand this comment. In our revision, we decided to bring up the term "Brownian motion" only when we define the stochastic process, commonly known as Brownian motion (top of page 6 of revised manuscript).

11. *Page 5, line 18: Concerning a constant D, see Meduri & Wicht (2016).*

    We have added clarification of this issue.

12. *Page 5, line 22: . . . well potential potential .*

    We fixed this typo (and many more).

13. *Caption of fig. 2: ... and potential U(x), with U?(x)=- v(x) ... Or use an integral formulation.*

    This figure has been removed in the revision because it was not essential.

14. *Page 6, line 4: Could you explain iid in a few words for the non-experts?*

    We added a definition of iid.

15. *Page 6, line 9: ... are affected by affected by ...*

    We fixed this typo (and many more).

16. *Caption of fig. 3: There is something wrong with the sentences.*

    We fixed the issues in the figure caption. Figure 3, however, has changed during our revision (we made it into two separate figures).

17. *Page 8: Because an SDE is noisy ... Well, no surprise there. Could you be more specific? Wouldn't this also depend on the noise parameters? How expensive is an SDE integration? How long do you have to integrate?*

    We added clarification of this issue. The Reviewer is right in that the required simulation length depends on the noise parameters (small noise will not be an issue). With nominal parameters we decided that even a simulation of 10 billion years is not sufficiently accurate (in the resulting PSD) for our purposes. Our sampling approach requires repeated simulations. One MCMC run requires about 1 million simulations. We perform six different MCMC runs to check the validity of error models, the model's limitations and the impact of the various data sources on parameter estimates. Even with a 10 billion year simulation time, this would require a little more than 1 month or so in computation time (not using parallelism in the MCMC, timing a 10 billion year simulation at 3.6 seconds and assuming $10^6$ simulations for the MCMC). The code that uses the approximations runs in less than a day. We have also validated our approximations against long simulations (using only nominal parameters).

18. *Page 8, line 10: The comparison does not look too good. Please discuss.*

    We added a discussion of the model-data fit and its quality.

19. *Page 8, line 14: ... the time averaged value of the absolute value of x(t) ..*

    This sentence does not appear in the revised manuscript.

20. *Page 8, line 23: approximating $\rightarrow$ approximate*

    This sentence does not appear in the revised manuscript.

21. *The SD is not very close to the observation. Please discuss.*

    We added a discussion of the standard deviation not being very close to the observation.

22. *Sampled once per year . . . (see above).*

    See above.

23. *Discuss the comparison.*

    We added a discussion of the comparison.

24. *... the bound may be overly pessimistic ... Well, the problem is not solved yet, but it looks like electron-electron interaction could at best only have a mild effect. Anyway, there is no need to dive into this topic in the paper and I would simply drop this sentence.*

    We have revised our derivation of the bounds according to the Reviewer's suggestion.

25. *Note that the stochastic model represents longer statistical time scales. Arguing with flow velocities is of limited use here.*

    The only place we mention flow velocity is where we put bounds on the parameter $\gamma$. We simply require the effects of diffusion to be small relative to the induction term, which depends on velocity. We make no claim that the stochastic model provides an estimate of flow velocity.

26. *Sections 4.2.2 and 4.2.3: Please provide more details here. (See comment above)*

    We have added new explanations and supplied more details in our revision.

27. *Figure 6: Provide colorbar. Discuss a bit more. How well are the parameters constrained, for example compared to the priors?*

   The use of a color-bar is unusual in the context of corner plots. We added explanations of how the posterior distributions compares to the uniform prior based on the bounds in Section 3.4.

28. *Page 17, line 2: The "however" seems out of place.*

   We revised this sentence based on the Reviewer's suggestion.

29. *Page 17, line 5: … the impact of each …*

   We fixed this typo.

30. *Page 21, line 1: … in the context of our simplistic model CALS10k.2 mostly constraints …*

   We revised the sentence based on the Reviewer's suggestion.

31. *Page 21, line 30: Every geomagnetic data point is indeed the result of hard work, but why is this a challenge for the model described here?*

   We agree, this is not an "issue" here and we removed this part of the sentence. The hard work required for each data point contributes to the fact that the amount of data we have is rather limited.

32. *Page 22, top: Incorporating different type of "data" in a Bayesian approach is a standard application. Please point out the specific novelty in your approach.*

   We revised our explanations to emphasize why our overall approach is useful. To be sure, we do not "invent" new algorithms in this paper, we apply known numerical techniques to an interesting and important problem and discuss their uses and, to some extent, their implications.

33. *Page 22, line 4: "We use the full paleomagnetic record"? The years of hard work have resulted in more than just Sint-2000 and PADM2M.*

   We agree, we revised this statement according to the Reviewer's suggestion.

Schmitt, D., Ossendrijver, M.A.J.H. and Hoyng, P., Magnetic field reversals and secular variation in a bistable geodynamo model, Phys. Earth Planet. Int., 125, pp. 119-124,

**Response to Reviewer 2**

*The authors take an interesting Bayesian approach, but there is a number of issues which require an improvement of the manuscript.*

We thank the Reviewer for their comments. We have thoroughly revised the manuscript throughout to address the concerns. We have expanded explanations and (hopefully) clarified all the issues that are brought up.

*The developed stochastic model uses of paleomagnetic and archeomagnetic data. How do the authors treat uncertainties in dating, smooth the data and carry out other data massages?*

The treatment of uncertainties is discussed in Section 4.2. The validity of the assumptions about our error models (uncertainties) are assessed by a suite of numerical experiments in Section 6. Uncertainties in dating are not important for our purposes (assuming that the overall length of the Sint-2000 and PADM2M time series is about 2 Myrs and the overall length of CALS10k.2 is about 10 kyrs). We do not smooth the data or carry out any other data "massages". We use the Sint-2000, PADM2M and CALS10k.2 data sets directly, without modifications.

*How can one combine many frequency-domain data points with a few time-domain data points?*

We discuss this issue in Section 4.2.1 and in Section 6. We "artificially" decrease the error covariances of the few time-domain points to increase their impact on the parameter estimates. We clearly spell out the consequences this action has on the resulting uncertainties of posterior estimates. A related issues was brought up by Reviewer 1, who also suggested an alternative. We explained in our response to Reviewer 1 that we do not anticipate that the suggested alternative will lead to improved results.

*Please, discuss the geomagnetic power spectra and the respective models in some more detail. How do they incorporate them in the Bayesian approach?*

This issue was also brought up by Reviewer 1. The power spectral densities are computed from the time series (Sint-2000, PADM2M, CALS10k.2) using the multi-taper technique of Constable and Johnson (2005). Sections 4.2.2 and 4.2.3 describe how the PSDs are incorporated in the Bayesian approach (via feature-based likelihoods).

[revised manuscript text omitted]

$$
\rho = \begin{array}{c|ccccc}
 & \bar{x} & D & \gamma & T_s & a \\
\hline
\bar{x} & 1.00 & 0.78 & 0.20 & 0.02 & -0.03 \\
D & 0.78 & 1.00 & 0.64 & 0.02 & -0.03 \\
\gamma & 0.20 & 0.64 & 1.00 & -0.01 & -0.02 \\
T_s & 0.02 & 0.02 & -0.01 & 1.00 & 0.00 \\
a & -0.03 & -0.03 & -0.02 & 0.00 & 1.00
\end{array}
\tag{27}
$$

5   The strong correlation between $\bar{x}$ and $D$ and $\gamma$,  is due to the contribution of the reversal rate to the overall likelihood (see Equation (20)) and the dependence of the spectral data on $D$ and $\gamma$ (see Equations (9) and (10)). From the samples, we can also compute means and standard deviations of all five parameters and we show these values in Table 4.

The table also shows the reversal rate and VADM standard deviation that we compute from  2,000 samples of the posterior distribution  followed by evaluation of Equations (20) and (13) for each sample. We note that the

10 reversal rate (4.06 reversals/Myr) is lower than the reversal rate we used in the likelihood (4.23 reversals/Myr). Since the posterior standard deviation is 0.049 reversals/Myr, the reversal rate data are about four standard deviations away from the mean we compute. Similarly, the posterior VADM standard deviation (mean value of  $1.77 \cdot 10^{22}$ Am$^2$) is also far , as measured by the posterior standard deviation, from the value we use as data ($1.6610^{22}$ Am$^2$). These large deviations indicate an inconsistency between the VADM standard deviation and the reversal

15 rate. A higher reversal rate could be achieved with a higher VADM standard deviation, The reason is that the reversal rate in Equation (20) can be re-written as

$$r \approx \frac{\gamma}{2\pi} \cdot \exp\left(-\frac{\bar{x}^2}{6\sigma^2}\right) \cdot 10^3 \; \text{Myr}^{-1}, \tag{28}$$

using $\sigma \approx \sqrt{D/\gamma}$, i.e., neglecting the correction factor due to sedimentation, which has only a minor effect. Using a time average of $\bar{x} = 5.23 \cdot 10^{22}$ Am$^2$, a reversal rate $r = 4.2$ reversals / Myr, setting $\gamma = 0.1 \text{kyr}^{-1}$ (posterior mean value), and

20 solving for the VADM standard deviation results in $\sigma \approx 4.53 \cdot 10^{22}$ Am$^2$, which is not compatible with the SINT-2000 and PADM2M data sets (where $\sigma \approx 1.66 \cdot 10^{22}$ Am$^2$). It is possible that the low-frequency data sets underestimate the standard deviation and also the time average. For example, Ziegler et al. (2008) report a time average VADM of $7.64 \cdot 10^{22}$ Am$^2$ and a standard dev

The model fit to the spectral data  is illustrated in the left panel of Figure 7. Here we plot 100

25 PSDs we computed from 2 Myr and 10 kyr model runs and where each model run uses a parameter set drawn at random from the posterior distribution. For comparison, the figure also shows the PADM2M, Sint-2000 and CALS10k.2 data as well as $5 \cdot 10^3$ realizations of the low- and high-frequency error models. We note that the overall uncertainty is reduced by the Bayesian parameter estimation. The reduction in uncertainty is apparent from the expected errors generating a "wider" cloud of PSDs (in grey) than the posterior estimates (in blue and turquoise). We further note that the PSDs of the models, with

30 parameters drawn from the posterior distribution, fall largely within the expected errors (illustrated in grey). In particular the high-frequency PSDs (the CALS10k.2 range) are well within the errors we imposed by the likelihoods. The low frequencies of Sint-2000/PADM2M are also within the expected errors and so are the high-frequencies beyond the second roll-off due to the sedimentation effects. At intermediate frequencies, some of the PDSs of the model are outside of the expected errors. This indicates a model inconsistency in the frequency domain: based on the data (spectral *and* time domain) and the assumed error models, it is difficult 
[revised manuscript text omitted]
 good understanding of model limitations and inconsistencies further allows for selecting a different set of parameters for different "tasks" of the model. To study hypotheses about the reversal record, for example, one may want to de-emphasize the model fit to the spectral data (similar to configuration (a) or (d)). Any conclusions based on the model should then incorporate the fact that the model may not match other aspects of the field.

**8   Summary and conclusions**

15  We designed a Bayesian estimation problem for the parameters of a family of stochastic models that can describe the Earth's magnetic dipole over kyr and Myr time scales.  A main advantage of our approach is that it can be used to discover inconsistencies between

20  model and data. Once these inconsistencies are identified, several strategies can pursued to resolve these inconsistencies, e.g., improving the model or resolving consistency issues of the data themselves. The framework can also be used to reveal the impact that some of the individual data sets have on parameter estimates. The Bayesian approach we describe here leads to a stochastic SDE model with stochastic parameters whose distributions are informed by the geomagnetic data. Moreover, we can use our techniques to study the limitations and (remaining) uncertainties of the model. The result is thus a reliable,

[revised manuscript text omitted]

---

## Author Response (AR2)

**A comprehensive model for the kyr and Myr time scales of Earth's axial magnetic dipole field – response to Reviewers' comments**

Matthias Morzfeld, Bruce A. Buffett

We thank Reviewer 3 for their careful reviews and good suggestions. We incorporated most of the Reviewers' comments into our revised manuscript. A detailed, point-by-point response is provided below. We repeat the Reviewers' comments in italic and our responses appear in standard font. We also provide a latex-diff that can be used to track all changes from our original submission to this revised version. The line numbers below (and in the reviews) refer to the original submission.

**Response to Reviewer 3**

**Major comments**

*Bayesian inference can in principle be used to estimate, and quantify the uncertainty in, model parameters. The paper uses a MCMC sampler to perform this task to quantify the uncertainty in five parameters and complete a stochastic model for the earth's geomagnetic field. In particular, a recent approach is used in which certain 'features' are selected from the data, and a data model for the features is then constructed. The paper is well written and interesting, but I have several concerns that should be addressed. My largest concern has been raised already by the former reviewer: that the decision to reduce the time-data to three points, but to only reduce the spectral data to hundreds of points, has introduced unknown flaws to the inference algorithm. The authors partially address this issue by tuning the variance of the three time-data points, but this has artificially created a narrow posterior in several parameters. I do not think there is sufficient information to trust e.g. the estimate for the parameter a, which changes by two standard deviations when \*not\* narrowing the variance. This issue with the inference scheme is critical because the key contributions of the paper are all to the understanding of advantages and limitations of the chosen dipole model. But if the inference scheme is flawed, so is the inference, and we don't learn anything about the model...*

*I think one of several strategies should be adopted by the authors. Either the feature DA should be adjusted to remove the need for ad hoc tuning, or the tuning should be better justified, or the tuned results should come later in the paper. I have elaborated on these options in great detail (sorry) below.*

*1) adjustment: somehow the issue caused by low-dimensional, high variance time data should be resolved without resorting to ad hoc methods. As already suggested by a prior reviewer, the spectral data could be reduced to dimension 3-10, commensurate with the time-data. The reviewer suggests a parametrization of the spectral features. I think something along these lines should be tried. The paper already lays the foundation for one simple approach. Equations (24) and (25) describe the spectral 'feature observation operator' in three components corresponding to $f < 0.05$, $0.05 < f < 0.5$, and $0.9 < f < 9.9$. The high dimension comes from evaluating these components at many points, but really one has already made an assumption that the spectral data is split into three features. Perhaps one could replace the high-dimensional spectral data with these three features. The first feature would be (e.g.) the average distance from $y_{lf}$ to $h_{lf}(\theta)$ for $f < 0.05$; the second feature is the same thing but for $0.05 < f < 0.5$, and the third feature is again the mean innovation, but in the high-frequency regime. This choice reduces the spectral data to the same dimension as the time*

*data, and the appropriate covariance matrix can be calculated from the values already used in the paper.*

*2) If the tuning cannot be removed, then it needs to be discussed in detail. Obviously this was the goal of the authors, and I acknowledge the very relevant and interesting observations made from comparing the different observation configurations in Section 6. However, I think an analysis of the \*direct\* consequences of the variance tuning is missing - though much of the legwork has been done. -In which parameters is it likely that the posterior is too narrow? -What is the difference between configuration (e) and the (not shown) configuration that, like (e), only assimilates the time data, but that uses the correct variances for the data? -Is it certain that there no issue with degeneracy in the MCMC scheme due to the tight likelihood around the time data? (I note that MCMC Hammer is affine invariant, and therefore the answer to this question is likely in the affirmative; but I would like to see something on this in the paper.) Once this analysis has been done, the claims of the abstract and introduction must be re-evaluated in terms of this analysis, in particular "Our approach thus results in a reliable stochastic model for selected aspects of the long term behavior of the geomagnetic dipole field whose limitations and errors are well understood". At the present moment I would not say the limitations and errors are well understood; not by me, at least.*

*3) If the tuning cannot be removed or justified better, then please present configurations (d) and (e) first. Then configuration (a) can be framed as a (partially successful!) attempt to obtain consistent parameters from two inconsistent data sources, and this can all be achieved with minimal changes to the existing discussion. If the authors do not feel like my requests (1) or (2) are achievable, I think this is a very valid rearrangement of the manuscript. The chief issue with configuration (a) is that it is presented first, gets its own section, and is treated - in terms of manuscript structure - as the prominent method, with other methods present for discussion. If instead configuration (a) came later as a comparison, rather than as the figurehead scheme, then I have much weaker objections to the variance reduction.*

We thank the Reviewer for the suggestions. We decided to follow suggestion (2) – an expanded explanation of the consequences and limitations of our approach. We would also like to remind the Reviewer that a large part of Section 6 is devoted to this issue.

The reasons for following suggestion (2) are as follows. Option (1), i.e., reducing the number of spectral data to the order of the number of time data, is also ad-hoc and also requires tuning. There is no clear path as to how to reduce the number of spectral data points as is evident by two reviewers suggesting two different strategies. In view of these limitations, "tuning" the observation error covariances seems in fact cleaner than fiddling with the data because reducing the observation error covariance amounts to only one parameter (observation error covariance) being tuned and the consequences of this tuning are predictable (too sharp of a posterior). It is difficult to anticipate in what way a reduction of the spectral data would modify the shape of the posterior distribution.

Option (3), i.e., presenting results that do not show an overall good match with the data, is a matter of presentation. We prefer to show "the best" results (we could obtain with our approach) first and then discuss limitations.

*My second major contention is that the authors do not seem to mention investigating the validity of the Gaussian assumption for the features' likelihood(s). This assumption, while necessary for the approach taken, should be mentioned and (even if only briefly) investigated or justified with suitable references.*

A Gaussian assumption for observation errors is widely used, but we agree with the Reviewer that there are other options. We added a few sentences in the revision to clarify our assumptions and

mention that other error models are also possible.

**Minor comments**

*p1 abstract: "Another important aspect of our overall approach is that it can reveal inconsistencies between model and data..." You have identified inconsistencies; please strengthen the statement accordingly.* Thank you, we revised the abstract accordingly.

*p.3 line 7: were→where*

Thank you, we fixed this typo.

*p.3 line 15, "For example, a point-wise mismatch of model and data is difficult to enforce when two different data sets report two different values for the same quantity": I am not convinced of the statement as written. If I have a 1d state $x$ and two observations $y_1$, $y_2$ of the same state variable (here just $x$), then (assuming obs errors are independent) the observation operator is $H = [1; 1]$ and the covariance matrix is $R = [\sigma_1\, 0; 0\, \sigma_1]$. That is, there is no issue with multiple observations of the same quantity, in principle. I think the issue here is the unknown amount of observation error, or bias, means that the two observation sets are very inconsistent, and that this bias is impossible to model (?).*

The Reviewer is correct: the situation is more complicated than our one-sentence-description suggests. We decided not to bring up delicate details in the introduction and removed the statement about observing the same quantity in various ways because this is not essential to the points we want to make.

*p3, line 25: either have 'likelihood' or 'likelihoods' for both*

Thank you, we fixed this typo.

*p9, last line: "This will lead to an improved fit, along with an improved understanding of model uncertainties." just repeats the previous sentence*

Thank you, we removed this sentence.

*p13, Section 4.2. The Feature-based likelihoods introduction repeats much from the introduction/motivation sections. Consider removing the first paragraph, and/or perhaps the last few sentences from p.13*

Using features is one of our main points and we decided to repeat this point so that the reader is reminded of our overall approach.

*p15 "...but alternative strategies are straightforward to implement within our overall feature-based Bayesian estimation approach." This phrase will need updating one way or another. I recommend removing it no matter what happens to the rest of the paper. I got angry reading this because if a better strategy would have been straightforward to implement - why not do it??*

We apologize for upsetting the Reviewer. The anger was likely caused by the Reviewer reading quickly: we do not claim that "better" strategies are easy to implement, we suggest that *alternative* strategies are easy to implement. Nonetheless, we removed our statement.

*p23: "The configurations we consider are summarized in Table 4 and the corresponding parameter estimates are reported in Table 5." On my copy of the manuscript, the references are off by one (ie the configurations are summarised in table 5 and estimates reported in table 6).*

Thank you, we fixed this issue.

*p.29, code and data availability: If allowed by the journal, please clarify for the reader which repository/directory contains the appropriate code, or a starting example (e.g. suggesting the file to run to simulate the results for Figure 6), or upload a readme file to the repo. These are nice results and I think only a little work will make them more accessible.*

We include the directory where the code can be found.

[revised manuscript text omitted]